# In search of diverse and connected teams: A computational approach to assemble diverse teams based on members' social networks

**Diego Gómez-Zará**[1,2,3]\*, **Archan Das**[4], **Bradley Pawlow**[5], **Noshir Contractor**[5,6,7,8]

**1** Department of Computer Science and Engineering, University of Notre Dame, Notre Dame, Indiana, United States of America, **2** Department of IT, Analytics, and Operations, University of Notre Dame, Notre Dame, Indiana, United States of America, **3** Facultad de Comunicaciones, Pontificia Universidad Católica de Chile, Santiago, Chile, **4** School of Computer Science, Carnegie Mellon University, Pittsburgh, Pennsylvania, United States of America, **5** Department of Computer Science, Northwestern University, Evanston, Illinois, United States of America, **6** Department of Industrial Engineering and Management Sciences, Northwestern University, Evanston, Illinois, United States of America, **7** Department of Communication Studies, Northwestern University, Evanston, Illinois, United States of America, **8** Department of Management and Organizations, Northwestern University, Evanston, Illinois, United States of America

\* dgomezara@nd.edu

**Data Availability Statement:** We have deposited a minimal data set in the GitHub repository https://nusoniclab.github.io/. This repository contains the following files: (1) the pre-processed and de-

## Abstract

Previous research shows that teams with diverse backgrounds and skills can outperform homogeneous teams. However, people often prefer to work with others who are similar and familiar to them and fail to assemble teams with high diversity levels. We study the team formation problem by considering a pool of individuals with different skills and characteristics, and a social network that captures the familiarity among these individuals. The goal is to assign all individuals to diverse teams based on their social connections, thereby allowing them to preserve a level of familiarity. We formulate this team formation problem as a multi-objective optimization problem to split members into well-connected and diverse teams within a social network. We implement this problem employing the Non-dominated Sorting Genetic Algorithm II (NSGA-II), which finds team combinations with high familiarity and diversity levels in $O(n^2)$ time. We tested this algorithm on three empirically collected team formation datasets and against three benchmark algorithms. The experimental results confirm that the proposed algorithm successfully formed teams that have both diversity in member attributes and previous connections between members. We discuss the benefits of using computational approaches to augment team formation and composition.

## Introduction

Forming teams today is different from past decades. Nowadays, organizations and institutions aim to assemble groups based not only on members' expertise but also on diversity criteria [1, 2]. Because the workforce is becoming increasingly more diverse, more organizations are committing to bringing together members from different educational backgrounds, functional backgrounds, and demographic attributes in the same team [3, 4].

identified data used in this study, (2) the Python scripts to pre-process the original datasets, (3) the Python scripts that run the proposed algorithm and benchmark algorithms, including their plots and quantitative metrics. The pre-processed data in this repository constitutes the minimal dataset required to replicate all study findings reported in this article. The first dataset (MyDreamTeam) is administered by the SONIC Research Group, Northwestern University. We deposited a de-identified and minimal dataset from MyDreamTeam in the Github Repository to replicate the findings reported in this article. Because of the sensitive nature of some of the variables collected, Northwestern University Institutional Review Board (IRB)-approved protocol does not permit individual-level data to be made unrestricted and publicly available. Researchers interested in obtaining a larger sample of this restricted and de-identified individual-level data should contact the authors to inquire about receiving an IRB-approved institutional data sharing agreement. The second dataset used (bibsonomy) is administered by the Knowledge and Data Engineering Group, University of Kassel. This dataset is available under a license agreement, and it can be requested at https://www.kde.cs.uni-kassel.de/wp-content/uploads/bibsonomy/. While we do not maintain this dataset, we provide the scripts to generate the pre-processed datasets used in this study. The third dataset (GHTorrent) is administered by Georgios Gousios. The dataset is freely and publicly available at https://ghtorrent.org/. While we do not maintain these datasets, we provide the scripts to generate the pre-processed datasets used in this study.

**Funding:** This study was supported by the National Institute of Health (1R01GM112938-01, 1R01GM137410-01), the National Aeronautics and Space Administration (80NSSC21K0925), and the National Science Foundation (SMA-1856090) through grants awarded to NC. This study was also supported by the Directorate for Social, Behavioral and Economic Sciences (SES-2021117) and Microsoft Research (2020 Microsoft Research Dissertation Grant) through grants awarded to DG. The funders had no role in study design, data collection and analysis, decision to publish, or preparation of the manuscript.

**Competing interests:** The authors have read the journal's policy and have the following competing interest: DG received a research grant from Microsoft. This does not alter our adherence to PLOS ONE policies on sharing data and materials. There are no patents, products in development or

Numerous studies show the potential benefits of diversity in teams [5, 6]. At the identity level, research shows that demographic diversity—team members of different gender, culture, race, etc.—can boost team performance. Cultivating demographic diversity in teams can bring different traits, points of view, and experiences inherent to the demographic group [7, 8]. Some examples are gender diversity, which promotes productivity in software development teams [9], teams' collective intelligence [10], and innovations in R&D groups [11]. One study showed that racial diversity can also bring alternative perspectives and stimulate creativity, generating more original and competitive ideas [12]. Cultural diversity is another example: it helps teams produce more creative results than culturally homogeneous teams [13]. At the cognitive level, teams with high levels of functional diversity—that is, team members with different expertise, careers, and backgrounds—can deliver more original and creative outcomes. Promoting functional diversity can enhance creativity because it expands the breadth of information, knowledge, ideas, and perspectives within a team [14]. It also encourages divergent thinking, greater scope of skills, and idea recombination [15, 16]. As a result, functionally diverse teams are more likely to solve complex problems that require creativity and innovation than homogeneous groups [2, 8, 16]. Overall, the interplay of demographic and functional diversity plays a role in how team members' differences leverage their work and performance [7].

Despite the potential benefits of diversity in teams, research also shows that diversity is a "double-edged sword" [17]. Prior studies offer mixed, and even contradictory, results of the effects of diversity on teams [14, 17, 18]. While functional diversity can cause coordination problems and conflicts in a group due to differences in training and knowledge, demographic diversity can elicit inter-bias among members (i.e., "us-them" distinction) [19], leading to a lack of cohesion, communication, and trust [20–22]. For decades, organizations have promoted diversity training to help members work with others who are different from them. Although, when people are assigned to work in a diverse team, they are less likely to engage with the team and be motivated to work with teammates that differ in demographic or functional attributes [23].

One potential solution to moderate the adverse effects of diversity on teams is enabling team familiarity (i.e., team members' prior experience working with one another). A substantial body of literature shows that prior collaboration leads to a greater likelihood of success and future collaborations [24–26]. Team familiarity creates the foundations of trust, information distribution, and communication among members [27, 28]. And because team familiarity aids members in locating, sharing, and distributing their knowledge, team familiarity may address many problems created by diversity without compromising its potential benefits [29].

Can organizations assemble teams with high diversity levels and familiarity simultaneously? Rather than forming teams based on either diversity criteria or prior relationships, combining both can help members promote trust, and organizations make the benefits of diversity more salient [29]. In this work, we propose a computational approach to discover suitable team combinations that maximize team diversity and familiarity at the same time. We chose these two team characteristics because both can be determined during the team formation process. Since this task requires assessing all the possible combinations among the available members, we elaborate on an optimization problem and its algorithm implementations to find invaluable team combinations efficiently.

We formulate this team formation problem as a multi-objective optimization problem to assemble teams maximizing their diversity and familiarity simultaneously. We use Harrison and Klein's framework [30] to calculate teams' diversity based on the variety and disparity of attributes, and we use Kargar and An's communication cost metric [31] to calculate teams' familiarity based on members' social network structure [32]. We then implement this problem

marketed products associated with this research to declare.

employing the *Non-dominated Sorting Genetic Algorithm II* (NSGA-II). This implementation is appropriate because it provides a set of efficient team combinations and considers the trade-offs of different objectives. We demonstrate the effectiveness of our approach using three datasets that contain team membership information: (1) students self-assembling teams using the *MyDreamTeam* platform [33], (2) scientists co-authoring papers provided by the *bibsonomy* dataset [34], and (3) teams collaborating on GitHub provided by the *GHTorrent* dataset [35]. We assess our proposed algorithm against other multi-objective optimization methods highly cited in the literature by evaluating its solutions and running time. The results demonstrate that our proposed algorithm successfully provided solutions with higher diversity and familiarity levels.

The main contribution of this paper is the formulation of the team formation problem considering teams' diversity levels and members' familiarity simultaneously. While most studies in team formation algorithms have considered members' skills or personal costs as team formation objective functions [36], we formulate this optimization problem based on different operationalizations of diversity (i.e., disparity and variety of attributes). The second contribution of this work is the design of algorithms for this team formation problem that assigns all available individuals to a team. Previous team formation problems have mainly focused on finding the best team from a pool and dismissed the remaining individuals [36, 37]. This work also provides theoretical implications for team research. In particular, the use of computational mechanisms to support the team formation processes [38–40]. Practical implications of this study contribute to several communities invested in increasing team diversity. Since team builders cannot solve this problem quickly by manually checking each team combination, algorithms can automatize this task by bringing together members who possess existing social connections while, at the same time, from different backgrounds, characteristics, and expertise levels [41, 42]. Expanding the use of this algorithm to broader audiences could provide new benefits for groups that seek to embrace diversity and keep high familiarity levels.

This article is an extended and revised version of a preliminary conference proceeding presented in Complex Networks 2020 [43]. Compared with the conference article, this version (a) presents a review of team formation algorithms, (b) extends the definitions and pseudo-codes of the proposed team formation problem and algorithm, (c) upgrades the proposed algorithm to handle isolated individuals and when the number of available individuals is not a multiple of the team size, (d) evaluates the algorithm with three datasets to prove that our optimization problem can work in other team formation domains, (e) compares its performance against other benchmark multi-objective algorithms, (f) uses quantitative metrics to compare the algorithms' results, (g) elaborates on the findings and implications of this work for researchers and practitioners, and (h) provides the scripts to pre-process the datasets, the pre-processed datasets, and the scripts with our proposed algorithm and benchmark algorithms for reproducibility purposes.

## Related work

Computer science scholars have elaborated different approaches to solving the problem of team assignment [36, 44, 45]. Scholarship has concluded that finding the most efficient team combinations from a pool of individuals is a challenging computational problem, and it is even harder for individuals to solve manually. It is a complex task that requires assessing all the possible combinations among the members of a pool, which can become an insurmountable combinatorial challenge. Given a pool of $n$ members that must be assigned into teams of size $k$, we must calculate an iterative permutation where we can select the first $k$ members from $n$, then other $k$ members from $n - k$, and so on. Assuming that $k$ is a multiple of $n$, we

have to calculate $n/k$ permutations. As a result, we must compute $n!/(k!^{n/k*}(n/k)!)$ possible team combinations for a pool of $n$ members. If we want to assemble teams of size three from a class of 18 students, there are 190,590,400 possible combinations ($18!/(3!^{6*}6!)$). Assessing all these combinations scales in factorial time (i.e., $O(n!)$). Therefore, this task cannot be done in polynomial time and demands different approaches to find solutions efficiently.

Recent literature reviews [36, 37, 46] characterize team formation algorithms according to three main dimensions: (i) the number of teams that result from the algorithm, (ii) the members' attributes considered by the algorithm, and (iii) the number of objective functions considered by the algorithm.

## Number of teams

Most solutions propose to find the best team possible from a given pool of individuals. The "best-team" approach usually considers the team formation problem as an assignment problem, where the goal is to find the best members who can assemble a team. Key contributions to this literature rely on methodology innovations. For example, El-Ashmawi et al. [47] searches for the team with the least communication costs among team members using an implementation of the particle swarm optimization algorithm. Bhowmik et al. [48] developed a team formation algorithm using a submodular function optimization. This implementation finds the best team of experts with relaxed constraints: teams "must" have some skills while they "should" have others. Lastly, Keane et al. [49] employs a team formation algorithm using a gradient boosting framework to find the minimal team with experts who can work effectively together. A limitation of these methods is that they provide only a single "best" team rather than multiple teams that include all the members from the available pool.

A few studies have explored the problem of assigning all available individuals to teams. One approach is forming multiple teams through iterative heuristics. In this case, teams are assembled by extracting $k$ members of the pool according to an objective function until no more members are left without a team. One example is Agrawal et al. [50], which proposed heuristic algorithms to maximize the gain (or minimize the cost) aggregated over all the teams assembled from a pool of available individuals. This paper presents two iterative heuristic algorithms that team up "strong" members with others who are "weaker" than them. As a result, experts are distributed among several teams. A second approach is formulating the team formation problem as a partition problem. A pool of individuals is partitioned into teams using heuristic metrics for all the assembled teams. Some implementations use clustering algorithms that aim to find members sharing similar characteristics. Some examples are Nurjanah et al.'s implementation [51] that uses Fuzzy C-Means to cluster individuals in homogeneous teams [51], and Srba and Bielikova's implementation [52] that clusters students according to specific collaborative characteristics. A third approach is finding efficient team combinations using evolutionary algorithms [53, 54]. In a nutshell, evolutionary algorithms start assigning all members to random teams and then alter individuals' memberships iteratively to find better team combinations. After evaluating the combinations using determined objective functions, evolutionary algorithms keep the best team combinations to find new combinations in the next iteration. One example is Agustín-Blas et al. [53], who developed a genetic algorithm that organizes individuals into groups and searches for team combinations that maximize groups' required resources.

## Members' attributes

The second dimension focuses on the members' attributes considered by the algorithm. The algorithms' goal is to find members that maximize specific team attributes, such as the number

of social connections among members or the number of skills covered by the team. Most algorithms set the presence of *skills* (or expertise) in a team as the main goal. An illustration of this approach is Zakarian and Kusiak's algorithm [55], which uses mathematical programming to find members that contribute a particular skill to the team. Other computational implementations propose to assign members according to their *roles*. An algorithm based on this approach is Yannibelli et al. [56], who developed an evolutionary approach that assembles student teams by assigning members to specific roles. Moreover, algorithms can include members' *social networks* when forming teams. Lappas et al. [57] searches for the best team with the lowest social distance among members (i.e., a surrogate for communication costs). This article proposes two algorithms called "Rarest First" and "Enhanced Steiner." Given the problem of assembling the best team of size *k* with members who have the skills to solve a task *T* from a social network *G*, the first algorithm searches for the smallest graph diameter possible among *k* members who have the skills to solve *T*. The second algorithm searches for the smallest subset of edges that connects *k* members with the skills to solve *T* (i.e., minimum spanning tree). Both algorithms will aim to find the best team possible given a particular social network. Other implementations consider the personnel cost, members' availability, and workload balance among members as part of the team formation problem [58].

## Number of objective functions

The third dimension is the number of objectives being optimized by the team formation algorithm. Some examples are minimizing teams' communication costs, minimizing teams' personnel costs, and maximizing the number of skills present in each team. Most algorithms define the team formation problem with a single objective with restrictions [59]. The examples mentioned before follow this single-objective function design. A pitfall is that other beneficial goals for team composition cannot be considered during the optimization process simultaneously (e.g., minimizing communication costs while maximizing the team's skills).

Prior studies have introduced more than one objective function to the team formation problem. One example is Kargar et al. [60], which presents the "Minimal Cost Contribution" algorithm (MCC). Its goal is to search for the team with the lowest communication costs and the lowest personnel costs simultaneously. MMC's objective function is a linear combination of both cost functions with a parameter λ that indicates the trade-off between communication and personal costs. This algorithm implements a heuristic approach that adds new members to the team incrementally and considers the costs of adding a new member with respect to the current costs of the assembled team. Despite the benefits of these linear combination formulations, this approach presents two limitations: it provides only one single team solution, and its trade-off variable for the cost functions must be set in advance. Thus, finding other suitable solutions using these methods depends on the adjustment of the trade-off variable, which can add bias to the search process [61].

Recent algorithmic contributions have formulated the team formation problem as a *multi-objective* optimization problem to optimize two or more objective functions simultaneously [62, 63]. These problems involve trade-offs between two or more objectives since improving a solution in one objective is possible only by making a concession to another objective. Thus, multi-objective optimization problems do not provide a single solution but obtain multiple solutions considering different relevance emphases for the multiple objectives. While in single-objective optimization problems the superiority of one solution over others is determined by the objective function, in multi-objective optimization problems it is determined by dominance. The optimization process looks for solutions that are better than others in all the objective functions. As a result, the problem delivers a set of "non-dominated" solutions, which

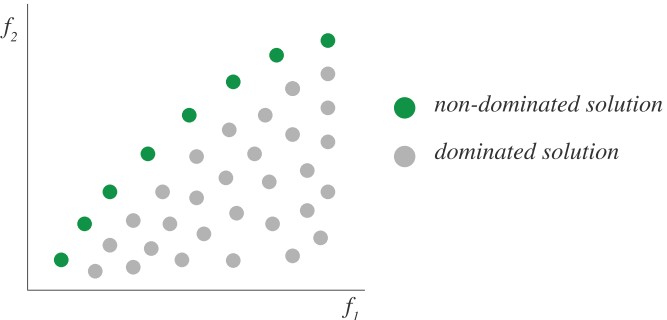

**Fig 1. Pareto front example.** Non-dominated solutions represent optimal solutions considering trade-offs between the two objective functions.

consists of solutions that can be improved without a simultaneous detriment to at least one of the other objectives. Multi-objective optimization is also known as Pareto optimization. Fig 1 shows an example of a Pareto front showing different non-dominated solutions ranging between two objectives. Computing this Pareto front allows decision makers to compare and check different trade-offs among both dimensions.

Based on this approach, multi-objective algorithmic implementations provide a set of team solutions that consider varied assessments of the objective functions [54, 64]. Zhang and Zhang's implementation [64] selects the members with the highest capabilities for the task and the best interpersonal relationships to assemble the best team. This study uses the particle swarm optimization implementation to determine whether a member *i* must be part of the best team. Solutions move in a two-dimensional continuous space, and the algorithm applies a sigmoid function to binarize members' presence. Perez-Toledano et al. [63] developed a genetic algorithm to find competitive basketball teams considering the cost and valuation of each player simultaneously. Each solution consists of a team from a set of available players, and its final Pareto front displays different teams that consider the trade-off between players' valuation and cost. Based on these formulations, team builders can see and compare other teams and choose what objective they will prioritize when selecting a team.

## Problem formulation

After reviewing relevant team formation problems and their respective algorithms, we aim to implement this particular problem that maximizes teams' diversity and teams' familiarity simultaneously. This problem is appropriate for a multi-objective optimization formulation since maximizing teams' familiarity could lead to forming groups with members that are similar to each other [65]. Although we could implement this problem as a single-objective optimization problem, we would have to prioritize one of those goals and avoid trade-offs among solutions. Moreover, prior formulations of the team formation searched for either the best team among multiple objectives or team combinations based on a single objective. We propose a multi-objective optimization problem that assigns all available individuals into teams, resulting in several team combinations that consider different relevance emphases for diversity and familiarity. This work is not the case for previous studies on team formation and provides a new approach to the team formation literature.

## Materials and methods

In this section, we introduce the multi-objective problem and definitions that we will use throughout this paper. Our notation is also summarized in Table 1. We also describe the

**Table 1. Notation.**

| Variable | Definition |
|---|---|
| $P$ | Set of people |
| $n$ | Number of people |
| $C$ | Set of categorical attributes |
| $U$ | Set of numerical attributes |
| $m$ | Number of categorical attributes |
| $l$ | Number of numerical attributes |
| $G$ | Social network |
| $p_j$ | $j$th person |
| $d(p_i, p'_j)$ | Distance between persons $i$ and $j'$ |
| $T$ | Set of teams |
| $q$ | Number of teams |
| $k$ | Number of members per team |
| $\mathcal{P}$ | A set of team solutions (i.e., Population) |
| $r$ | Number of Team combinations (i.e., Population size) |
| $F$ | Pareto front |

NSGA-II implementation of this multi-objective problem and its components. We then describe the datasets and benchmark algorithms we used to evaluate the team formation problem. Finally, we explain the quantitative metrics to compare algorithms' results.

## Definitions

**Members, attributes, networks, and teams.** We consider a set of participants $P = \{p_1, p_2, \ldots, p_n\}$ with a set of categorical attributes $C = \{c_1, c_2, \ldots, c_m\}$ and a set of numerical attributes $U = \{u_1, u_2, \ldots, u_l\}$. These individuals' attributes have different scales and represent information about each person (e.g., age, gender, race, skill). Depending on the individuals' information available, teams can have several attributes describing their qualities and composition. Each person has a value in each one of these attributes. We denote $c_i(p_j)$ to obtain the value of the categorical attribute $c_i$ for the person $j$. Similarly, we use $u_i(p_j)$ to obtain the value of the numerical attribute $u_i$ for the person $j$. Person $j$ can be represented as a vector of these categorical and numerical attributes. Thus, we have the attributes of $p_j$ as $(c_1(p_j), \ldots, c_m(p_j), u_1(p_j), \ldots, u_l(p_j))$.

People are connected together in a social network modeled as an undirected and unweighted graph $G$. We define $G = (P, E)$, where $E$ represents the graph's edges. Each node in $G$ represents a person from $P$. We use person and node interchangeably throughout this paper. Two people are connected by an edge if they have collaborated in the past. In other words, if individuals $i$ and $j$ have worked together, then $G_{i,j} = 1$. Otherwise, $G_{i,j} = 0$.

Given this list of participants $P$ connected in the network $G$, the goal is finding a set of teams $T = \{t_1, t_2, t_3, \ldots, t_q\}$, where all members of $P$ assemble $q$ teams and belong to one team only. The optimization dual-problem can be formulated as minimizing the communication costs among team members and maximizing teams' diversity levels. We now make these notions and describe each objective function.

**Communication costs.** Lappas et al. [57] focused on the importance of collaborations and familiarity between experts by considering the cost of their collaborations. According to this model, experts who collaborated together in the past are more likely to exchange information and ideas effectively than experts without prior collaborations. Based on experts' prior

collaborations, this model calculates the *communication costs* among team members to estimate their collaboration and familiarity levels. The goal of optimizing communication costs is to form teams with high familiarity levels. A literature review shows that communication costs are a highly used proxy for collaboration and familiarity among researchers [66].

In our setting, we use communication costs as a proxy for teams' familiarity. Kargar and An [31] found the total *sum of distances* between team members to be a reasonable measure of communication costs, as it is more stable to changes in the network than other potential measures. Other alternatives for communication costs are the social network's diameter (i.e., the largest shortest path between any two nodes in the network), and the minimum spanning tree (i.e., the minimum sum of the weights of a network's edges) [57]. We also implemented this problem using these two definitions, and their results were similar to those obtained using the sum of distances. The results of the diameter's implementation are available in S1 Fig and S1 Table in S1 File, and the results of the minimum spanning tree's implementation are available in S2 Fig and S2 Table in S1 File.

We define the communication costs between two individuals $p_i$ and $p_j$, denoted as $d(p_i, p_j)$, as the shortest path length while traversing the edges of the graph $G$ from one node to another. If $p_i$ and $p_j$ have collaborated in the past, they are at one-hop of distance. If $p_i$ and $p_j$ have not collaborated together but have a prior collaborator in common, they are separated by two-hops. Having common past collaborators within a team can promote familiarity based on "triadic closure" [67]. This mechanism posits that nodes are more likely to establish a new connection when they have a connection in common. Three-hops and 4-hops can follow the same principles based on "balance mechanisms" [67]. Individuals will tend to forge new connections with collaborators of their collaborators to seek consistency within their group. Therefore, using the total sum of distances in our objective function aims to search for teams that maximize the number of direct collaborations (i.e., one-hops), common connections (two-hops), and close connections (three-hops or higher). The lowest communication cost value is when all team members have collaborated with each other (i.e., they are directly connected), and the highest is when team members are not connected at all. In this implementation, if there is no path between $p_i$ and $p_j$ in $G$, we set the communication costs between them as the social network's diameter.

We define the communication costs of a team $t$ as the total sum of shortest path lengths between members, as it is more stable to changes in the network than other potential measures. We denote by $Cc(t)$ the communication costs of team $t$, which has $k$ members. Thus, we define the team $t$'s communication costs as:

$$Cc_t = \sum_{i,j \in t, i \neq j}^{k} d(p_i, p_j) \tag{1}$$

The goal is to minimize the average sum of shortest path lengths across all assembled teams in the individuals' network. Computing the sum of communication costs of a set of teams runs in $O(n^2)$ time.

**Team diversity score.** The second goal is to generate diverse teams with a broad array of backgrounds, traits, and skill repertoires. Diversity describes the distribution of differences among the members of a unit with respect to a common attribute [30]. Harrison and Klein [30] presented a framework suggesting that diversity is best conceptualized in three ways: separation, variety, and disparity. Separation refers to differences among team members in their lateral position on a continuum (e.g., value, attitude, belief). Variety refers to categorical differences among team members wherein the number of represented categories contribute to team diversity (e.g., gender, career, race). Finally, disparity represents differences in the

concentration of valued assets or desirable resources (e.g., expertise, educational level, tenure). These metrics allow researchers to operationalize functional and demographic diversity in parallel and according to their theoretical conceptualizations [14].

In this implementation, we use variety metrics to assess teams' diversity provided by $C$ categorical variables, and disparity metrics to assess teams' diversity provided by $U$ numerical variables. To measure each team $t$'s variety metrics by its members' categorical attributes $C$, we use the Blau index ($B_{t,c_i}$) [30]. This index quantifies the probability that two team members randomly selected from the team would be in different categories. A low score means members fall into the same category, whereas a high score means members fall into different categories. We denote $p_{c_{ij}}$ is the proportion of members who fall into a particular category $j$ in the categorical attribute $c_i$. Given that the number of categories in $c_i$ is $o_{c_i}$, where $j = 1, ..., o_{c_i}$, the Blau Index' formula for the team $t$ is:

$$B_{t,c_i} = 1 - \sum_{j}^{o_{c_i}} p_{c_{ij}}^2 \qquad (2)$$

To measure each team $t$'s disparity metrics by its members' numerical variables $U$, we use the coefficient of variation ($CV_{t,u_i}$) [30], which is defined as the ratio of the standard deviation to the mean of the attribute $i$, $u_i \in U$. A low coefficient of variation score means that all team members $t$ have similar levels of the attribute, whereas a high score means all team members $t$ have different levels of the attribute. For a team $t$ with members $j = 1, 2, \ldots, k$, and with $\bar{u}_i$ as the team's mean value of the attribute $i$, the formula is the following:

$$CV_{t,u_i} = \sqrt{\sum_{j \in t}^{k} (u_i(p_j) - \bar{u}_i)^2 / k} / \bar{u}_i \qquad (3)$$

These two team diversity measures are useful because they do not change when the input data is scaled linearly, and they both tend to stay around the same values. Given that the team formation problem considers $C$ categorical variables and $U$ numerical variables, the diversity measures can be weighted to prioritize differences within a specific variable. The vector of weights $W$ has $|C| + |U|$ elements, where $W = (w_{u_1}, ..., w_{c_m}, w_{c_1}, w_{c_2}, ..., w_{c_l})$. Based on these measures, we aggregate diversity for different attributes into a single value. We define the team diversity score $V$ of a team $t$ as the weighted sum of the Blau indexes for all $C$ categorical variables and the coefficient of variation for all $U$ numerical variables. The formula is:

$$V_t = \sum_{i=1}^{m} w_{u_i} * CV_{t,u_i} + \sum_{i=1}^{l} w_{c_i} * B_{t,c_i} \qquad (4)$$

**Multi-objective problem statement.** We formulate the problem as a multi-objective problem with the goal of finding a set of $r$ team solutions $\mathcal{P} = \{T_1, T_2, T_3, ..., T_r\}$, where each $T$ represents a potential solution with $q$ teams. The decomposition of the assessment function into both objectives—minimizing communication costs and maximizing team diversity score—allows us to find multiple solutions that the single-objective approach cannot reach. As a result, we expect to find not a unique solution $T$ but a set of solutions $\mathcal{P}$ for which there is not another feasible solution better in both objective functions. This set of solutions $\mathcal{P}$ is also known as a Pareto front, where (a) there exists no other set of solutions $T'$ with more diverse and connected teams, and (b) each solution $T_i, i \in \mathcal{P}$ is not superior to all the other solutions

in $\mathcal{P}$ with respect to both diversity and communication costs objectives. Having this set of team solutions $\mathcal{P}$ makes it possible to assess each of them individually, so a team builder can choose the most appropriate teams possible to assemble for the given context and circumstances.

In sum, the team formation problem addressed in this paper is to find the Pareto front $\mathcal{P}$ of team solutions, where each solution $T$ is composed by $q$ teams ($T = \{t_1, t_2, t_3, \ldots, t_q\}$). The dual objective is maximizing teams' diversity based on the categorical attributes $C$ and numerical attributes $U$ and minimizing the communication cost based on $G$. We can model this problem as:

$$\min \quad Cc(T) = \sum_{t=1}^{q} Cc_t;$$

$$\max \quad V(T) = \sum_{t=1}^{q} V_t;$$

$$\text{s.t.} \quad t \subseteq P$$

$$\sum_{i=1}^{q} |t_i| = n$$

$$t \in T; \forall t = 1 : q$$

(5)

Since finding teams from a graph $G$ while minimizing the sum of shortest path lengths and team allocation problems is proven to be an NP-hard problem [57, 68], this multi-objective problem is also an NP-hard problem.

## NSGA-II implementation

Pareto fronts' shapes provide helpful information about the degree of trade-off between different objectives and how much compromise is needed from some criteria to improve others. Determining the exact Pareto front for multi-objective combinatorial optimization problems is difficult since it is necessary to compute all the possible combinations to find the true Pareto front [63]. For this reason, the goal is to find an approximation of the true Pareto front using heuristic algorithms. A critical assumption for these algorithms is that the Pareto front is sufficiently populated. The quality of this approximation depends upon (1) the proximity of the points on the approximated front to the points on the true Pareto front; and (2) the diversity of the solutions on the approximated front, where more diversity is typically better. Although the true Pareto front is unknown, solutions that dominate others are close to the theoretical true Pareto front. Therefore, the diversity of the solutions will provide a more extensive range and granularity of the Pareto front.

Genetic algorithms (GA) are commonly used to find approximations of Pareto fronts [69]. By imitating evolution in nature, this method optimizes a population of initial solutions into better solutions through natural selection. Each solution is characterized as a *chromosome* (i.e., a vector of attributes), which can be mutated and altered in each iteration. The best solutions will endure after they mutate over time. Genetic algorithms are ideal for finding solutions for optimization problems in large and highly non-linear spaces [70].

The genetic algorithm starts from a population of randomly generated solutions evolving into new solutions through an iterative process. The population created in each iteration is also known as a *generation*. In each generation, the algorithm evaluates each population's chromosome according to the objective function in the resolved optimization problem. The

| Team 1 | | | | Team 2 | | | | | Team $q$ | | |
|---|---|---|---|---|---|---|---|---|---|---|---|
| 3 | 25 | 17 | 32 | 15 | 32 | 53 | 21 | ⋯ | 5 | 10 | 45 | 31 |

**Fig 2. Team chromosome representation.** Members only can be part of one team.

chromosomes with the highest scores are selected from the current generation and used to form a new generation. This process continues until a maximum number of iterations is achieved or by a threshold function defined for the solutions.

We implemented a genetic algorithm called *Non-dominated Sorting Genetic Algorithm-II* (NSGA-II) formulated by Deb et al. [71]. NSGA-II allows finding an approximation of the Pareto front, having different team solutions $\mathcal{P}$ that variate according to the communication costs and diversity score specified. The NSGA-II approach is based on sorting the populations into a hierarchy of sub-populations using Pareto dominance criteria. Then, chromosomes for the next iteration are selected according to the mentioned hierarchy. This *elitist* selection guarantees that potential good chromosomes are kept in the population, and the solution quality obtained does not decrease from one iteration to the next. The solutions are also ordered according to the similarity among their chromosomes, removing redundant ones to promote diversity in the Pareto front. As a result, NSGA-II can converge on a high-performing Pareto front after a few iterations. Previous work has shown that NSGA-II provides solutions with high levels of efficiency running in $O(n^2)$.

In this implementation, each population $\mathcal{P}$ contains $r$ team solutions $\mathcal{P} = \{T_1, T_2, ..., T_r\}$, and each chromosome represents a potential set of $q$ teams $T_i = \{t_1, t_2, \ldots, t_q\}$. We use "chromosome" and "team solution" interchangeably throughout this paper. We characterize a chromosome as a vector of individuals partitioned into $q$ parts to obtain the teams (Fig 2). As a result, each chromosome's length equals the number of people $n$, representing $q$ teams of size $k$ ($q^*k = n$). We adapted this algorithm to our specific diverse team formation problem, and we outline these steps in Algorithm 1.

**Algorithm 1**: NSGA-II scheme pseudo-code

```
Input: Population size r, People P, Number of teams q, Number of Gen-
       erations g
Output: Solutions 𝒫
𝒫 ← ∅
for i ← 1 to r do
  Shuffle (P)
  T ← Split(P, q)
  Add T to 𝒫
for i ← 1 to g do
  // Generate offspring
  Children ← ∅
  for j ← 1 to r do
    p₁← RandomChoice (𝒫)
    p₂← RandomChoice (𝒫)
    Add CrossoverAndMutation (p₁, p₂) to Children
  // Merge the parent population with the children population
  𝒫 ← Merge (𝒫, Children)
  // Evaluate communication cost and diversity score
  EvaluateCommunicationCost (𝒫)
  EvaluateDiversity (𝒫)
  // Perform non-dominated sort
  F← FastNonDominatedSort (𝒫)
  // Create a new population
```

```
𝒫' ← ∅
// Add fronts until we have the allowed population size
k ← 0
while Size (𝒫') + Size (F_k) ≤ r do
  Add F_k to 𝒫'
  k ← k + 1
// Calculate the crowding distance of the last front
CrowdingDistance (F_k)
// Sort front's solutions according to their crowding distance
SortFronts (F_k)
// Select final chromosomes and add them to the new population
Δ ← r − Size (𝒫')
FinalChromosomes ← SelectFinalChromosomes (F_k, Δ)
Add FinalChromosomes to 𝒫'
// Update population for the next generation
𝒫 ← 𝒫'
return 𝒫
```

**Initialization.**   The algorithm starts by initializing a population of chromosomes $\mathcal{P}$ having teams assembled randomly. Its input parameters are the total number of chromosomes $r$ to include in the population $\mathcal{P}$, the list of people $P$, the number of teams $q$ to form, and the number of iterations to perform $g$. Chromosomes are stored as two-dimensional arrays of shape ($q$, $k$), where $q$ is the number of teams possible to assemble, and $k$ is the number of members per team. Each chromosome is a potential solution to the diverse team formation problem, and the goal is to find a set of chromosomes with high levels of diversity and low communication costs. After the initial population is created, the algorithm creates the offspring and searches for the Pareto fronts iteratively until the maximum number of generations $g$ is reached.

**Crossover step.**   In each generation, the algorithm takes two random chromosomes ($p_1$ and $p_2$) from the existing population $\mathcal{P}$ and randomly selects $q$ teams from this union. As a result, the algorithm will have a child chromosome with $q$ teams. Since the child's teams are randomly selected from two different chromosomes, individuals may get selected twice, coming from $p_1$ and $p_2$. The algorithm replaces repeated individuals with others who were not assigned to a team. It explores each member of the child chromosome and counts how many times an individual is part of a team. If an individual is counted more than once, this individual is randomly replaced by a missing member. At the end of this revision process, the algorithm will have the child chromosome with all the members of $P$ assigned to one team. These random samplings provide sufficient mutation for the algorithm to introduce diversity into the population without adding another mutation step. We outline the proposed crossover method in Algorithm 2.

**Union.**   After the crossover step, the algorithm combines the population $\mathcal{P}$ with its offspring, doubling the population's size (i.e., $2r$). The algorithm then calculates the diversity score $V$ and communication costs $CC$ of each chromosome of this union.

**Fast non-dominated sort step.**   Next, the algorithm must select the best $r$ chromosomes from this union of size $2r$. To find this set, the algorithm performs a *non-dominated sorting* among all the existing chromosomes from $\mathcal{P}$. The goal is to identify solutions that perform better than others and classify them according to their performance in different Pareto fronts $F$. The algorithm first checks the dominance relationships among all the chromosomes. Given two chromosomes, $T$ and $T'$, $T$ dominates $T'$ if and only if $Cc(T) \leq Cc(T')$ and $V(T) \geq V(T')$ with at least one strict inequality. In other words, $T$ is at least as good as $T'$ for all objectives and strictly better for at least one. This dominance relation is denoted as $T \prec T'$. If one of the objectives of $T$ is not better than $T'$, and it cannot be improved in value without degrading some of the other objective values, then $T$ is non-dominated by $T'$. One example of a non-dominated

solution is *T* having higher diversity scores but higher communication costs than *T'*. In that non-dominance case, either *T* and *T'* are feasible solutions for the next generation.

Once the algorithm maps all the chromosomes' dominance relationships, it creates a first Pareto front of solutions consisting of all the non-dominated solutions ($F_1$). This set is also denominated as the Pareto optimal. Then, the algorithm creates a second front of Pareto optimal solutions ($F_2$) that were disregarded in the first front, and so on. As a result, the algorithm sorts the population's chromosomes into a hierarchy of sub-populations. The sort keeps finding successive Pareto fronts until all chromosomes are assigned to a Pareto front.

**New population.** The algorithm then selects the best *r* chromosomes for the next generation. At a given time, there are 2*r* chromosomes sorted in the hierarchical Pareto fronts *F*. The algorithm creates the new population $\mathcal{P}'$ adding the chromosomes stored in the Pareto fronts. If the total size of the first Pareto front is smaller than *r*, then the algorithm adds all the chromosomes of this front to $\mathcal{P}'$. Then, the algorithm adds the remaining solutions for the new population from the subsequent non-dominated fronts. The algorithm continues this procedure until it can not add more fronts to $\mathcal{P}'$.

**Crowding distance.** The algorithm must add chromosomes to the new population until there are exactly *r* chromosomes. If the last selected non-dominated Pareto front $F_k$ has more chromosomes than the allowed to add to $\mathcal{P}'$, the algorithm must choose a smaller set from $F_k$ to complete the *r* chromosomes. Let $\delta = r - Size(\mathcal{P})$, the number of missing chromosomes to complete *r*. The algorithm identifies the best $\delta$ chromosomes from this last front $F_k$ by calculating the *crowding distance* among the chromosomes. This metric determines how similar the chromosomes are in terms of performance in the multi-objective problem. After calculating this distance, the algorithm ranks the chromosomes according to their distances and eliminates chromosomes that perform similarly to other chromosomes. This procedure keeps a broader front of solutions and removes redundant chromosomes. Then, the $\delta$ best chromosomes from $F_k$ are added to $\mathcal{P}'$. As a result, $\mathcal{P}'$ counts with the *r* best chromosomes and becomes the parent of the next generation, starting a new iteration.

**Output.** After the optimization runs through the previously specified number of generations *g*, the algorithm returns an approximation of the Pareto front $\mathcal{P}$ having *r* team solutions.

**Algorithm 2**: Crossover Function

```
Input: Parent p₁, Parent p₂, People P, Number of Teams q
Output: Children
p ← Concatenate (p₁, p₂)
Children ← AssembleRandomTeams(p, q)
MissingMembers ← Set (P) - Set (Children)
Counted ← ∅
for Child in Children do
  if Child in Counted then
    NewMember ← SelectRandomMember (MissingMembers)
    Replace Child ← NewMember
    Remove NewMember from MissingMembers
  else
    Add Child to Counted
  end
end
return Children
```

## Data

In this section, we evaluate the proposed algorithm for our team formation problem using three real-world datasets. The data sources are *MyDreamTeam* (a team formation platform), *Bibsonomy* (a social bookmarking site), and *GHTorrent* (a GitHub repository database). Using

**Table 2. Description of the MyDreamTeam, Bibsonomy, and GHTorrent datasets.**

| Case | Description | TS | N | E | D | MSD | De. | Cent. |
|------|-------------|----|----|----|----|------|------|-------|
| MDT Case A | An undergraduate course at an university in the US | 5 | 55 | 130 | 7 | 3.03 | 0.06 | 0.14 |
| MDT Case B | A graduate course at an university in the US | 6 | 61 | 518 | 4 | 2.23 | 0.14 | 0.42 |
| MDT Case C | An MBA course at an university in the US | 3 | 65 | 87 | 8 | 3.51 | 0.03 | 0.06 |
| Bibsonomy Case A | Papers published at *Nature* | 5 | 40 | 398 | 3 | 1.39 | 0.26 | 0.22 |
| Bibsonomy Case B | Papers published at *Science* | 5 | 100 | 2,238 | 4 | 1.14 | 0.23 | 0.24 |
| Bibsonomy Case C | Papers published at *Physica A*. | 5 | 86 | 160 | 4 | 1.58 | 0.02 | 0.07 |
| GHTorrent Case A | Repositories with *Python* as the main language | 5 | 57 | 318 | 4 | 1.64 | 0.10 | 0.12 |
| GHTorrent Case B | Repositories with *Java* as the main language | 5 | 100 | 4,683 | 2 | 1.05 | 0.47 | 0.03 |
| GHTorrent Case C | Repositories with *Ruby* as the main language | 5 | 55 | 642 | 3 | 1.17 | 0.43 | 0.22 |

We include *TS*: team size, *N*: the number of individuals, *E*: the number of relationships among individuals, *D*: the collaboration network's diameter, *MSD*: individuals' mean short distance, *De.*: the collaboration network's density, and *Cent.*: centralization score using degree centrality.

these datasets to simulate teams for this team formation problem illustrates the effectiveness of our framework in real scenarios. We show summary statistics from these datasets in Table 2. The resulting data and the scripts to pre-process the raw data are available at http://nusoniclab.github.io/.

**MyDreamTeam dataset.** We evaluate our proposed algorithm using data from real team formation cases. We extracted this dataset from the *My Dream Team Builder* [33], a recommender system to help individuals self-assemble teams. This dataset contains cases of participants self-assembling their teams. Cases date from 2014 to 2020. On this recommender system, participants create profiles, search for teammates, and send invitations to form teams. The cases consist of classes from universities in the United States. The dataset includes participants' traits, demographics, and social networks, which they reported in an initial survey. We selected three cases to test our algorithm: an undergraduate course, a graduate course, and an MBA course. Participants used the system to assemble teams for small group discussions.

Permission to collect data from participants was approved by Northwestern University Institutional Review Board (#STU00078513). All applicable institutional and governmental regulations concerning the ethical use of human subjects were followed during this research. Electronic consent was obtained from study participants via an online survey instrument. Participants were asked for their consent to use data collected through *My Dream Team Builder* for research purposes. We hashed users' identifiers to create a de-identified dataset.

**bibsonomy.** The second dataset is extracted from *bibsonomy* [34], a social-bookmarking and publication-sharing system. We chose bibsonomy since prior team formation papers tested their algorithms using this database [58]. This dataset is administered by the Knowledge and Data Engineering Group, University of Kassel. The bibsonomy dataset is available under a license agreement, and it can be requested at https://www.kde.cs.uni-kassel.de/wp-content/uploads/bibsonomy/. This dataset contains a large number of computer science related publications. Each publication is written by a group of authors. The bibsonomy website is visited by many users who use tags to annotate the publications. Following the procedure described by Anagnostopoulos et al. [58], we used the tags associated with each author's papers to represent their skills. Each author's skill represents the number of papers published with their respective tag. We selected three journals related to social network analysis to test our algorithm: "Nature", "Science", and "Physica A: Statistical Mechanics and its Applications." We counted the frequency of the tags in each of these journals and selected some popular tags related to our study. For the first two journals, we selected papers that included the tags *'network'*, *'social*

*network'*, and *'small world.'* Then, we identified the authors of these articles, created the co-authorship network, and selected authors from the largest component. Similarly, we did this procedure for the third journal using the tags *'network'*, *'graph'*, *'model'*, and *'system.'* We hashed the authors' names to create a de-identified dataset.

**GHTorrent.** We used GitHub data provided by the GHTorrent project [35], an offline mirror of the data offered through the GitHub API. This dataset can be downloaded at https://ghtorrent.org/downloads.html. The GHTorrent dataset covers a broad range of development activities on Github, including repositories, pull requests, and users. We downloaded the dataset dump "06/01/2019" to build our testing dataset. We filtered users who contributed between 40 and 80 projects to keep median users in our analysis. Following an approach similar to the bibsonomy dataset, we used programming languages associated with each user's contributed repositories to represent users' skills. Each user's skill represents the number of contributed projects written in a specific language. Since repositories can have files in multiple languages, we selected repositories' most used language as the repository's language. We selected three of the most popular languages in this dataset: Java, Python, and Ruby. Then, we identified the users of these repositories and created the collaboration network. In this example, users have a tie if they contributed to the same repository at least two times. Finally, we selected users from the largest component. We hashed the authors' names to create a de-identified dataset.

## Evaluation

We compare the proposed algorithm for the team formation problem (denoted as NSGA-II) against three well-known multi-objective optimization methods used for benchmark purposes [62, 72]:

**Pareto Local Search (PLS) method.** This iterative algorithm starts with a set of random solutions as the initial population and explores each solution's neighbors [73, 74]. The algorithm updates the population based on Pareto dominance: it will add non-dominated neighbors to the population and remove existing solutions that are dominated by the newly added solutions. Once the neighborhood of a solution has been fully explored, the solution is marked as explored. The algorithm iteratively explores new solutions as they are added to the population until no better solutions are found. After all the solutions are explored, and no more non-dominated solutions can be discovered, the algorithm stops. We implemented the version proposed by Zihayat et al. [72] for combinational problems. In this implementation, a solution's neighbors are all the possible team combinations from the solution with two members swapping teams. Since PLS does not depend on a fixed number of generations, we only run one iteration of this algorithm to compare its results with the other methods. Given $n$ individuals, and that the algorithm will explore $\binom{n}{2}$ neighbors of each solution, the computational complexity of this implementation is $O(n^3)$ in the best-case scenario.

**Strength Pareto Evolutionary Algorithm 2 (SPEA-2).** Like NSGA-II, this algorithm is based on elitist selection and dominance criteria [75]. Instead of creating different Pareto fronts, SPEA-2 keeps the set with the best solutions found in each iteration called "archive," which is separated from the population. The algorithm starts with random population solutions and an empty archive. Then, it calculates a fitness value for each solution based on (a) the number of solutions it dominates (i.e., strength), (b) the number of solutions by which it is dominated by the current population (i.e., raw fitness), and (c) its distance with other solutions (i.e., density value). The best solutions will be copied to the archive. After initiating the first population, the goal is to identify non-dominated solutions for the next generation. Based on the fitness values, the algorithm performs binary tournament, crossover, and mutation steps with the solutions from the current population and archive. These new solutions will

constitute the next population. After these processes, the algorithm checks how many non-dominated solutions result from the union of the current population and archive. If the number of non-dominated solutions is less than the archive's size, the archive will include some dominated solutions from the union. The algorithm selects dominated solutions based on their fitness values. If the number of non-dominated solutions is higher than the archive's size, the algorithm removes redundant solutions based on their nearest neighbor Euclidean distance. The next iteration will create a new generation based on this updated archive. We implemented the version proposed by Zitzler et al. [75]. We used the same number of generations from the NSGA-II testing and set the archive's size to equal the population's size. In the best-case scenario, the computational complexity of this algorithm is $O(M^2 log M)$ where $M$ is the sum of the population size ($n$) and archive size ($n'$).

**Hybrid Particle Swarm Optimization (HPSO) method.**　This algorithm combines the steps of particle swarm optimization algorithms (PSO) and genetic algorithms (GA) [76]. In its original version, PSO starts with a population of candidate solutions (called particles) and moves them around in the search space over the particle's position and velocity. Each particle's movement is influenced by its local best-known position, but is also guided toward the global best-known positions in the search space. In each iteration, the algorithm updates particles' positions based on their velocity. After a few iterations, the algorithm provides solutions that are approximations of local optima and global optima. Since the PSO's original formulation only operates in continuous optimization problems, we require a version that can handle combinational optimization problems. Moreover, PSO operates with a global optimum that does not exist in Pareto front problems. Zhang et al. [76] proposed a hybrid version that replaces the PSO's particle position and velocity update formulas with the genetic algorithm's crossover and mutation operations. In a nutshell, the HPSO algorithm iteratively examines each particle and (a) applies the crossover step with a random non-dominated solution found by the particle, (b) applies the crossover step with a random non-dominated solution known from all the population, (c) and performs the mutation step. If a resultant solution is better than the original, then the solution is updated. If a particle knows two or more non-dominated solutions, it will choose a random non-dominated solution as the best local particle. Similarly, if the population knows more than one non-dominated solution, it will select a random non-dominated solution as the best global particle. The running time of this algorithm is expected to be polynomial since it will check the $n$ solutions and run the crossover operation two times and the mutation operation once. As a result, the computational complexity is $O(n^2)$ in the best-case scenario.

We also compared the teams assembled by these four multi-objective algorithms with randomly assigned teams. Since the *MyDreamTeam* dataset already included fixed-size teams, we also computed the real teams' diversity scores and communication costs.

## Metrics

We computed the following quantitative metrics to evaluate the quality, quantity, and running time of the algorithms' solutions. These indicators map the final solutions to a number that indicates one or several aspects of the solution. We chose these metrics based on the literature review by Li et al. [77].

**Hypervolume ($HV$).**　This metric evaluates the total size of the objective space dominated by the algorithm's solutions with respect to a reference point. It can measure how close solutions are to the true Pareto front and how evenly spread the solutions are in the objective space. Algorithm $A$ will have higher hypervolume scores than algorithm $B$ if algorithm $A$'s solutions dominate algorithm $B$'s solutions. In this context, higher hypervolume scores show

that team combinations with higher levels of diversity and familiarity can be found. If the algorithm *A* finds team combinations with higher diversity scores and/or lower communication costs than algorithm *B*, the algorithm *A*'s hypervolume will be higher than the algorithm *B*'s hypervolume. The larger the *HV* value, the better the diversity and distribution of the team combinations. The *HV* of an algorithm **A** can be formulated as:

$$HV(\mathbf{A}) = \lambda(\cup_{\mathbf{a} \in A} \mathbf{x} | \mathbf{a} \prec \mathbf{x} \prec \mathbf{r}) \tag{6}$$

where **r** denotes the reference point, and λ indicates a measure to subsets of n-dimensional Euclidean space (i.e., Lebesgue measure). In our case, the hypervolume is the area of the rectangles formed by the solutions and a two-dimensions reference point.

**Unique Non-dominated Front Ratio (*UNFR*).**   This metric quantifies the contribution of each algorithm to the combined non-dominated front of all the algorithms. In this context, if algorithm *A* has a higher *UNFR* value than algorithm *B*, the former found team combinations with higher diversity and/or lower diversity scores than the latter. Let **A**$_\mathbf{unf}$ be the unique non-dominated front of a given algorithm **A**, then this metric is defined as:

$$UNFR(\mathbf{A}) = \frac{|\mathbf{a} \in \mathbf{A}_{\mathbf{unf}}, \nexists \mathbf{r} \in \mathbf{R}_{\mathbf{unf}} : \mathbf{r} \prec \mathbf{a}|}{|\mathbf{R}_{\mathbf{unf}}|} \tag{7}$$

where **R**$_\mathbf{unf}$ is the set of unique non-dominated solutions of the collections of all solutions produced by the algorithms. The *UNFR* value ranges from 0 to 1. An algorithm with a high *UNFR* value means that it contributed to many unique non-dominated solutions from all the non-dominated solutions found. In contrast, a value close to zero means that the algorithm provided a few unique non-dominated solutions to the final set.

**Computational complexity.**   Lastly, we evaluated these algorithms' computational complexity as a function of the input size. In this context, if algorithm *A* has a lower running time than algorithm *B*, the former can find team combinations from a pool of participants faster than the latter. Since some algorithms' running time can increase exponentially, this metric is relevant to measure how scalable and efficient the algorithm is when forming teams with large participant pools. We compared the algorithms' running times using different numbers of users from the GHTorrent "Java" and Bibsonomy "Science" datasets.

## Results

We ran the algorithms' evaluations for 50 generations with a population size of 50 chromosomes. We implemented these algorithms in Python 3.6.2. and performed the experiments on a server with a 2.60 GHz Intel(R) Xeon(R) CPU and 16GB of RAM. The algorithms' implementations and detailed results are available at http://nusoniclab.github.io/ for consultation. Table 2 shows the statistical data of the datasets, including the team size, the number of available individuals, the number of relationships, the diameter of the network, individuals' mean short distance, and networks' centralization.

Fig 3 shows the approximation of the Pareto front found by each algorithm in each dataset. The x-axis represents teams' total communication costs. Lower scores on this axis represent solutions with lower communication costs (i.e., teams internally more connected). The y-axis represents the total teams' diversity score of the solutions. Higher scores in that axis represent solutions with more diverse teams. As the results show, the NSGA-II implementation outperforms the benchmark algorithms in most of the tested datasets. NSGA-II found non-dominated solutions with high diversity values and low communication costs across all these databases. HPSO also contributed with non-dominated solutions to the final set of solutions. In particular, the plots show that HPSO was better at finding non-dominated solutions when

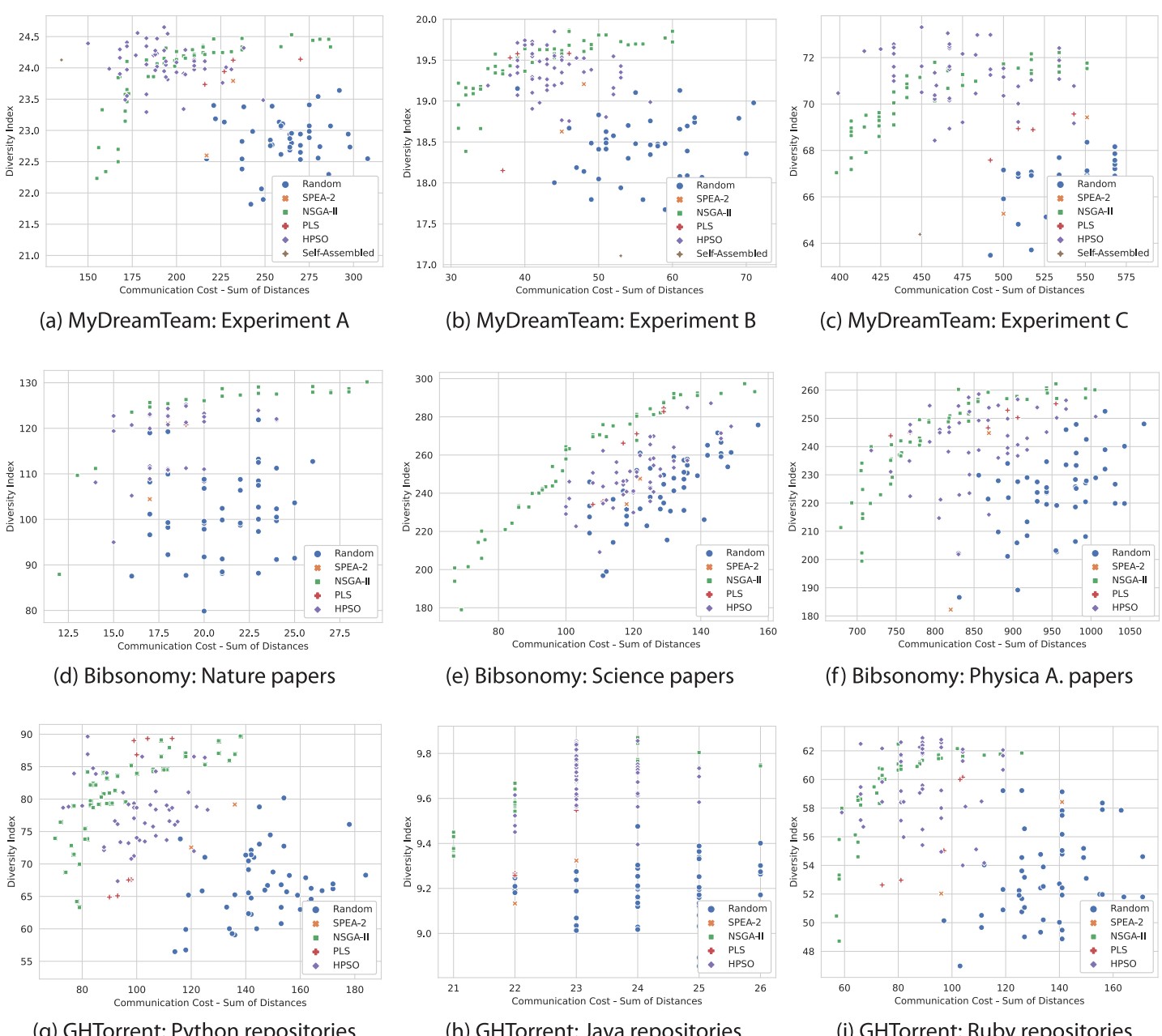

**Fig 3. Results of NSGA-II, SPEA-2, PLS, HPSO, and random assignment.** The NSGA-II implementation provided more diverse team solutions with high diversity scores and low communication costs than the solutions provided by PLS, HPSO, and SPEA-2. (a) MyDreamTeam: Experiment A. (b) MyDreamTeam: Experiment B. (c) MyDreamTeam: Experiment C. (d) Bibsonomy: Nature papers. (e) Bibsonomy: Science papers. (f) Bibsonomy: Physica A. papers. (g) GHTorrent: Python repositories. (h) GHTorrent: Java repositories. (i) GHTorrent: Ruby repositories.

setting a balanced trade-off between communication costs and diversity. Following NSGA-II and HPSO, PLS solutions were close and concentrated in certain regions of the team formation space. This concentration indicates that PLS tended to converge on certain non-dominated solutions, dismissing other potential team combinations that may not have been non-dominated in the first iterations. SPEA-2 results were worse than the other algorithms despite employing the same representation and operations. Overall, NSGA-II was better at finding

**Table 3. Hypervolume and unique non-dominated front ratio values for the five methods across the datasets.**

| | Hypervolume | | | | | Unique Non-dominated Front Ratio | | | | |
|---|---|---|---|---|---|---|---|---|---|---|
| | HPSO | NSGA-II | PLS | Random | SPEA-2 | HPSO | NSGA-II | PLS | Random | SPEA-2 |
| MDT Case A | **29,755** | 29,696 | 28,536 | 26,304 | 27,292 | **0.67** | 0.33 | 0 | 0 | 0 |
| MDT Case B | **30,298** | 29,972 | 29,151 | 28,506 | 28,593 | **0.86** | 0.14 | 0 | 0 | 0 |
| MDT Case C | 92,643 | **95,205** | 89,531 | 84,546 | 83,331 | 0.09 | **0.91** | 0 | 0 | 0 |
| Bibsonomy Nature | **368,697** | 364,369 | 339,493 | 345,935 | 322,730 | 0.29 | **0.71** | 0 | 0 | 0 |
| Bibsonomy Science | 684,658 | **721,970** | 674,118 | 631,915 | 529,396 | 0.00 | **1.00** | 0 | 0 | 0 |
| Bibsonomy Physics A | 386,482 | **394,852** | 356,903 | 321,826 | 294,007 | **0.44** | 0.33 | 0.22 | 0 | 0 |
| GHTorrent Case Python | 138,632 | **139,959** | 121,850 | 119,406 | 109,592 | 0.30 | **0.70** | 0 | 0 | 0 |
| GHTorrent Case Java | 15,572 | **15,614** | 14,967 | 14,923 | 14,886 | 0.33 | **0.67** | 0 | 0 | 0 |
| GHTorrent Case Ruby | 99,556 | **103,913** | 95,745 | 92,445 | 91,108 | 0.21 | **0.79** | 0 | 0 | 0 |

The best results are marked in bold.

solutions in the extremes of the approximate Pareto front, offering more variety of non-dominated solutions. It provided more alternatives compared to PLS, HPSO, and SPEA-2. Therefore, the NSGA-II implementation provides a spectrum of team solutions that team builders can explore and choose.

The results show that the NSGA-II algorithm achieved the largest hypervolume values on 6 of 9 datasets and the second-highest value for the other three datasets (Table 3). In other words, NSGA-II frequently found more team combinations with higher diversity levels and lower communication costs than the other algorithms. NSGA-II's high hypervolume values can be explained by its crowding distance step, which helped the algorithm find non-dominated solutions located on the extremes of the Pareto front. Since PLS and HPSO did not establish any criteria to avoid redundant solutions, their solutions resulted in a set of non-dominated solutions concentrated in certain areas. Therefore, the set of team combinations provided by NSGA-II frequently dominated the ones provided by the other algorithms.

The NSGA-II implementation also scored the highest unique non-dominated front ratio (*UNFR*) values on 6 of 9 datasets. In other words, NSGA-II frequently provided more non-dominated team combinations that the other algorithms could not find. A possible explanation for the NSGA-II second-place in the other cases is the low density in the collaboration network. Pools with few prior connections among individuals will reduce the number of possible highly-connected team combinations, making the heuristic search ineffective. In contrast, HPSO and PLS performed more crossover and mutation operations than NSGA-II. Executing these operations multiple times allowed HPSO and PLS to examine more team combinations and increase their likelihood of finding specific team combinations with low communication scores.

HPSO achieved the second-highest hypervolume and *UNFR* values. It benefited from the non-dominated solutions in the middle of the approximate Pareto front, which scored high diversity levels. These non-dominant solutions outperformed other algorithms and increased the area created by its approximate Pareto Front. SPEA-2 and PLS converged to a few solutions, covering a smaller area than the NSGA-II and HPSO solutions. Overall, NSGA-II found more non-dominated solutions across these two objectives and provided solutions with higher variance in communication cost values.

The large variance in both diversity and familiarity shows that the NSGA-II algorithm found more non-dominated solutions than the other algorithms, which is desirable to find the true Pareto front. The crowding distance step of NSGA-II allowed the algorithm to keep a

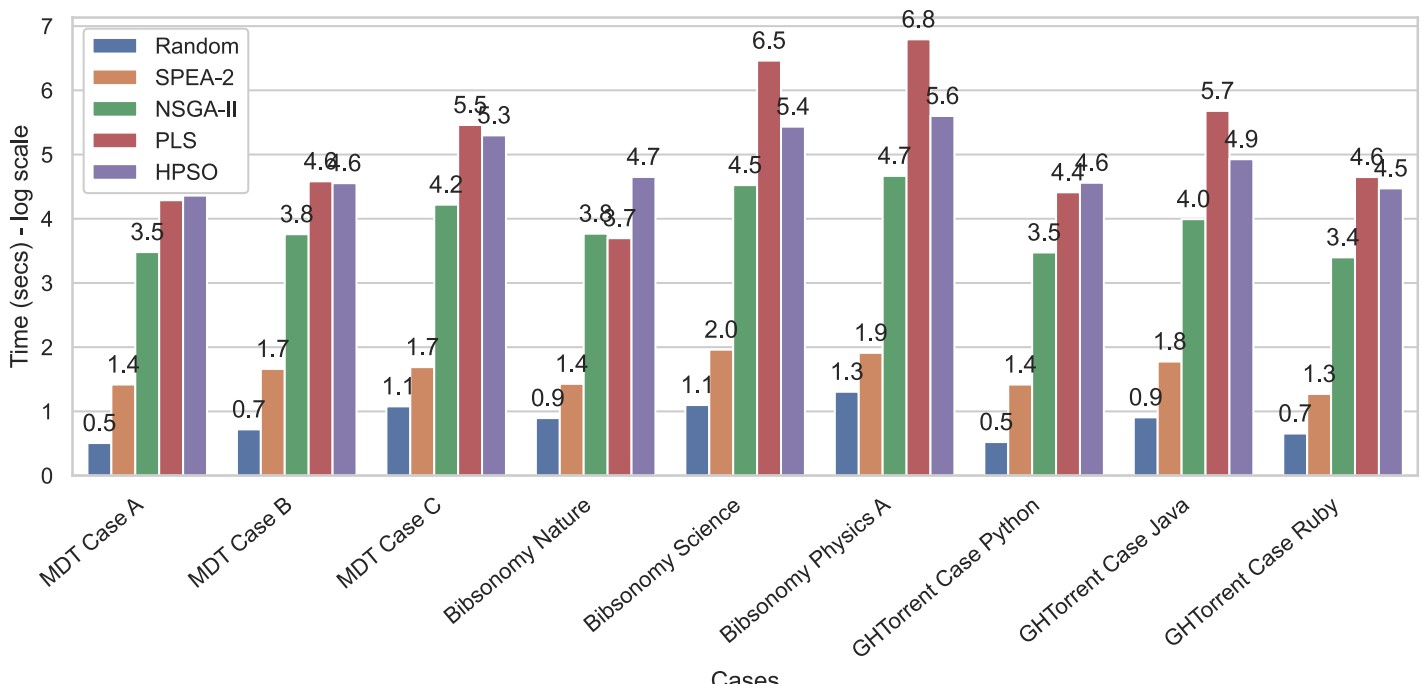

**Fig 4. Running time results for the NSGA-II, SPEA-2, PLS, HPSO, and random assignment.** All metrics are shown on a logarithmic scale.

broader range of non-dominated solutions. Plus, the algorithm kept secondary solutions in different layers that could have originated non-dominated solutions in later iterations. As the algorithm continues creating new generations, dominated solutions can still be considered to find other potential solutions. Moreover, NSGA-II could still identify non-dominated solutions in the middle of the trade-off. In contrast, the low variance of the other algorithms shows that they were likely to converge on a specific set of non-dominated solutions and to a specific trade-off. These algorithms did not include operations to diversify their current populations or remove redundant non-dominated solutions. Therefore, these algorithms could lack diverse solutions that reside in the extremes of the Pareto front.

Fig 4 presents the run time of all algorithms implemented. Fig 5 shows how algorithms' running time as a function of the number of available individuals. The results show that the NSGA-II implementation performs better than PLS and HPSO as the participant pool increases. PLS required more time to explore solutions' neighborhoods until all possible combinations were exhausted. In the case of HPSO, the two crossovers and one mutation step performed for each solution made the algorithm's operation three times longer than NSGA-II since the latter only performs one crossover step. Although HPSO took longer than NSGA-II, both worked in polynomial time ($O(n^2)$). Our results suggest NSGA-II required less than one-third of the time that PLS and HPSO took to provide similar results. Therefore, using NSGA-II is highly encouraged to find solutions efficiently as the input size increases. SPEA-2 did not find better solutions than PLS or NSGA-II, but its results converged faster than the NSGA-II and PLS methods.

Lastly, we calculated the frequency of direct contacts (1-hop), shared contacts (2-hops), 3-hops, and more in the assembled teams to understand the distances among team members (See S3 Table in S1 File). The results suggest that the vast majority of the members were connected with others through one intermediate (∼31%), followed by members who were directly

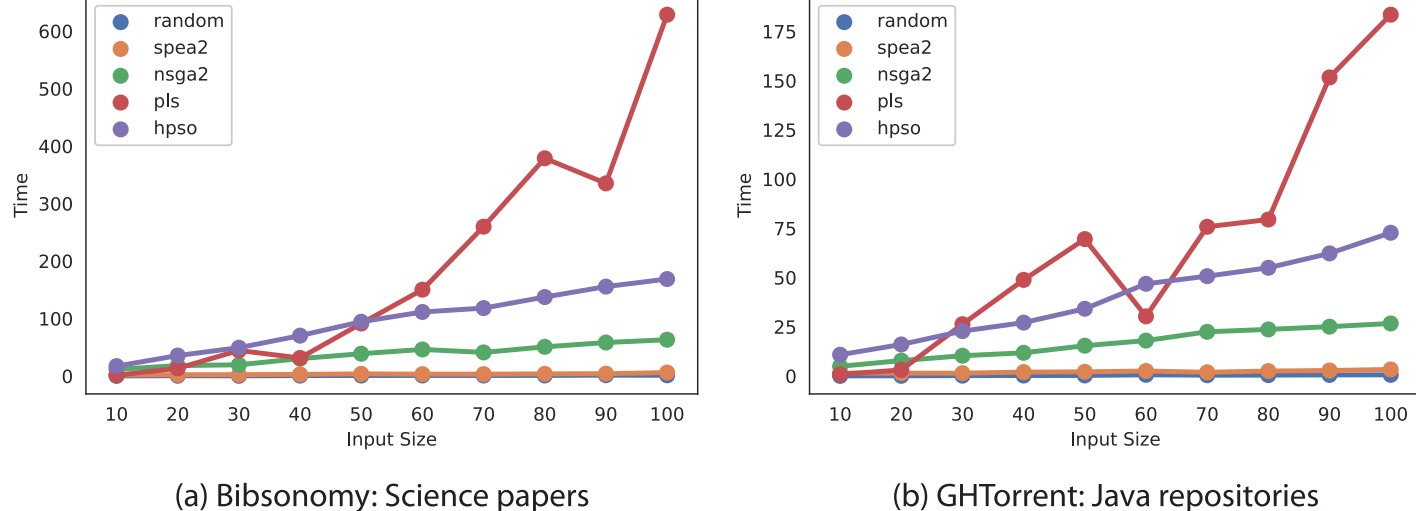

(a) Bibsonomy: Science papers    (b) GHTorrent: Java repositories

**Fig 5. Comparison of algorithms' time complexity by varying the number of users.** (a) Bibsonomy Science dataset. (b) GHTorrent Java.

connected (∼30%). These numbers show that the resultant teams were highly connected in general, and members connected through many hops were not representative.

## Discussion

Forming teams is a challenging task, especially when the goal is to bring a balance between diversity and members' teammate preferences. While prior work has focused on looking for the best team possible [37], the contribution of this paper is searching for balanced team combinations that distribute skills and connections equitably. Moreover, assembling teams that include both diversity in their members' contributions and prior relationships between members becomes an ultimate challenge to guarantee their success [78, 79]. In this work, we consider the problem of creating teams that are both diverse and highly connected teams using a computational approach. We implemented this team formation problem using a genetic algorithm that provides different team combinations according to teams' diversity and communication costs. The results indicate that diverse and highly connected teams can be assembled efficiently and quickly through this genetic approach. In the following subsections, we elaborate on the implications of this work and the potential applications.

This work shows the benefits of using computational approaches to assemble multiple teams that distribute members' skills among different groups and consider their prior relationships. Prior work mainly emphasizes finding the best team possible from a social pool (e.g., expert-team, all-star team) [36, 57, 58]. However, searching for multiple teams is also relevant in learning and organizational situations, such as forming student teams, forming operational teams within business units, or conducting scientific research. Because of the strong emphasis on assembling high-performing teams, algorithms that bring the best members together can create greater segregation within these social groups at the expense of others who are less skilled [80]. The concentration of skills and expertise in a small number of teams hinders the emergence of other teams having similar opportunities and social connections. As this paper demonstrates, the advanced computational infrastructure and the use of big data provide new opportunities to re-imagine several team combinations that individuals could not systematically and easily explore [37, 44]. Rather than creating teams using manual strategies or intuition, team builders can use algorithms to tailor different team combinations that optimize

diversity and familiarity simultaneously. Thus, implementations like the one presented in this paper allow team builders to create heterogeneous and diverse teams without sacrificing familiarity among team members, which is essential for successful collaborations [81].

Another contribution of this work is finding similar team combinations that individuals would assemble but with enhanced diversity levels. As prior studies have found, people tend to form teams with competent individuals and those who are familiar with them, enhancing the likelihood of satisfaction and commitment with the team [28, 82]. This fact is demonstrated in the *MyDreamTeam* dataset by comparing the lower communication costs of self-assembled teams and the higher communication costs of randomly generated teams. The proposed algorithm found team combinations with lower communication costs than self-assembled teams, suggesting that people have some intuition in forming well-connected teams. However, they lack reliable knowledge of higher-order connections among themselves. A possible explanation of this difference found by the algorithm is the tremendous challenge for individuals to discover and take advantage of indirect connections, such as shared contacts or shared past collaborators. Whether individuals assemble their teams or team builders design them, considering team members' indirect connections is not an easy task since indirect connections are not highly visible. In contrast, our algorithm excels in considering the broader social network structure given the global view of relationships between members. By using this algorithmic approach, individuals and managers can be more conscious of potential diverse teammates through their current relationships. Even if two team members do not know each other directly, teaming up with a shared "friend-of-a-friend" or indirect connections can potentially promote familiarity and psychological safety in teams [83–85].

Furthermore, we found that *MyDreamTeam* self-assembled teams were less diverse than the teams randomly generated by the algorithms. This tendency driven by homophily is consistent with prior literature, indicating that people prefer to team up with others who share similar characteristics [65]. Formulating this team formation problem provides new opportunities to boost team diversity over self-assembled teams while still considering high familiarity among team members. One main advantage of forming teams in this fashion is reducing individuals' biases. Since people naturally draw toward forming teams with similar individuals, an algorithm like the proposed one can augment people's decision-making process. Instead of connections driven by individuals' preferences, the algorithm can enact collective coordination by curating better team combinations that could satisfy individuals' expectations. This multi-objective approach can allow people to find feasible solutions that increase diversity without compromising familiarity in the team.

## Implications

This work provides theoretical implications for team research. In particular, the use of computational mechanisms to support the team formation processes. Literature has characterized team formation centered on behavioral mechanisms, where teams can be assembled by internal or external forces and based on similarity, familiarity, and competence [28, 86]. By formulating and implementing this multi-objective optimization problem, we found diverse and connected team combinations that individuals could not have foreseen. This work allows team scholars to reflect on the role of technologies in enabling new organizational structures among individuals and organizations, which could lead to new theories of team formation and the introduction of technologies [38–40].

Practical implications of this study contribute to several communities invested in increasing team diversity: managers assembling effective and diverse teams, instructors composing more diverse student teams, companies forming heterogeneous groups from different business

units, space agencies such as NASA forming composing space crews for long duration space exploration to Mars, and researchers investigating the use of algorithms for organizing scientific teams. Expanding the use of this algorithm to broader audiences can provide new benefits for groups that seek to embrace diversity and keep high familiarity levels. Furthermore, software developers and designers can use this study's implications for new procedures and guidelines for artificial intelligence in organizing workers. Finally, this work provides more computational approaches to enrich team formation processes [45, 87]. Since team builders cannot solve this problem quickly by manually checking each team combination, algorithms can automatize this task by bringing together members who possess existing social connections and, at the same time, have different backgrounds, characteristics, and expertise levels [41, 42]. We expect this work will assist in forming heterogeneous teams by considering diversity and social networks.

Another quality of this approach is adding more objectives to the team formation problem. For example, team builders could minimize other objective functions such as geographical distance among participants, personnel costs, or availability constraints. Likewise, this multi-objective problem can accommodate members' traits when diversifying is not desirable. As some prior meta-reviews indicate [14, 88], having a team with similar individuals may be desirable for low-difficulty tasks or when efficiency (rather than creativity) is the goal. Furthermore, it may be desirable for some traits such as personality or expertise to be similar rather than diverse [89]. This team formation problem can add another objective function that minimizes teams' diversity in some traits using the metrics defined by Harrison and Klein [30]. Therefore, one potential use of this algorithm is to maximize diversity in some members' attributes while minimizing diversity in other attributes.

Given this multi-objective approach's flexible trade-off, which solution should team builders consider from the Pareto front? Incorporating other metrics (e.g., individual performance, team cohesion, members' location) could help team builders select one specific team combination.

### Limitations and future work

It is important to acknowledge the limitations of this paper. First, the measures for diversity and communication costs were scaled specifically to each unique network and cannot be compared across different sets of participants. Second, the diversity measure is an aggregate of multiple diversity metrics for each attribute sampled; thus, it is difficult to assign any real meaning to the diversity metric apart from relative differences with the same network. Future implementations should consider how different diversity measures can be analyzed separately and according to the specific pool of participants. These might also weigh diversity on various dimensions or operationalize diversity metrics as different objective functions in the optimization problem. Third, forming scientific teams and software teams is more complex in reality: new members can be added over time, some specialization is required, not all of these teams share the same objectives, sizes, or restrictions, and diversity may be beneficial for only goals. We believe using the last two datasets should not be a concern because we use them only to test the algorithms' efficiency and results. This team formation algorithm can guide the formation of real scientific and software teams by finding more diverse and connected combinations. Fourth, we do not provide specific recommendations for demographic or functional diversity attributes. Prior studies have shown how the effects of diversity on team performance are mediated by contextual factors and team processes [14]. Team builders who want to administer this algorithm should reflect and decide on adding demographic and cognitive variables according to their organizational goals and particular context. Fifth, collecting social network data could be a big task for team builders. Assessing people's relationships can be done by

conducting surveys, examining communication networks, or tracing digital data [90]. Another potential strategy to build individuals' social networks is asking about their teammate preferences. The algorithm could find possible diverse team combinations based on individuals' responses [91]. Lastly, it is not possible to guarantee that the performance of the teams assembled by this algorithm will be better than other team formation strategies. Prior studies have shown mixed results for the direct effect of diversity on team performance in all contexts [14], as well as the advantage of using algorithmic approaches for team formation [92]. Other research has also shown that when individuals lack agency to self-assemble teams, they are less committed to their group [93, 94]. Future work should consider using this algorithm to assemble real groups and evaluate how well they perform compared to teams assigned randomly or by a manager.

Future work should add new restrictions to the multi-objective function, such as considering specific task roles in the teams, adding leaders to each team, or excluding certain team combinations in which some individuals do not want to work together. Using weighted networks could also provide more nuanced information about the strength of people's social relationships. One potential application is distinguishing individuals who have frequent interactions from those who barely speak to each other [95]. One example of potential areas of improvement is developing an automatic tuning for the weights assigned for each diversity attribute given a specific population. If the algorithm explores people's categorical and numerical attributes before conducting the team formation process, it could identify the attributes that have more variation and those that are scarce among individuals. Then, the algorithm could define the importance of each diversity attribute in the objective function. Lastly, the algorithm could be implemented as a web platform to provide more opportunities for managers, instructors, or researchers to assemble teams.

## Conclusion

This work addresses the problem of assembling teams from a social network that maximizes both diversity and familiarity. We formulated a multi-objective function for this problem and implemented a genetic algorithm to find well-connected diverse teams. In a thorough experimental evaluation, we evaluated the performance of our proposed algorithm and compared it against baseline approaches. We discussed the potential role of algorithms in augmenting team composition and helping team builders. In particular, computational approaches can be used to form teams that consider indirect connections and recommend combinations with higher diversity scores. As algorithms can discover more feasible team combinations than humans, team builders' decisions can become more structured, systematic, and comprehensive.

## Supporting information

**S1 File. Supporting figures and tables.** S1 Fig: Simulations using the Diameter metric. S2 Fig: Simulations using the Minimum Spanning Tree (MST) metric. S1 Table: Diameter Case. S2 Table: Minimum Spanning Tree Case. S3 Table: Team combinations' average proportion of hops.
(PDF)

## Author Contributions

**Conceptualization:** Diego Gómez-Zará, Archan Das.

**Data curation:** Diego Gómez-Zará.

**Formal analysis:** Diego Gómez-Zará, Archan Das.

**Funding acquisition:** Diego Gómez-Zará, Noshir Contractor.

**Investigation:** Diego Gómez-Zará.

**Methodology:** Diego Gómez-Zará, Archan Das, Bradley Pawlow.

**Project administration:** Diego Gómez-Zará.

**Resources:** Diego Gómez-Zará.

**Supervision:** Diego Gómez-Zará.

**Validation:** Diego Gómez-Zará.

**Visualization:** Diego Gómez-Zará.

**Writing – original draft:** Diego Gómez-Zará, Archan Das.

**Writing – review & editing:** Diego Gómez-Zará, Archan Das, Noshir Contractor.

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
