## [Decision Letter · Decision Letter 0]

31 Aug 2021

PONE-D-21-21409

In search of diverse and connected teams: a computational approach to assemble diverse teams based on members' social networks

PLOS ONE

Dear Dr. Gomez-Zara,

Thank you for submitting your manuscript to PLOS ONE. After careful consideration, we feel that it has merit but does not fully meet PLOS ONE’s publication criteria as it currently stands. Therefore, we invite you to submit a revised version of the manuscript that addresses the points raised during the review process.

As you can see below, despite being quite positive concerning your manuscript, both reviewers have asked for clarifications regarding a number of aspects. In particular, when revising your text, please pay special attention to the following issues:

- Clarify the contribution of this article beyond that of the conference paper mentioned in the text (as required by Reviewer 1). Notice PLOS ONE's publication criterion #1 (https://journals.plos.org/plosone/s/criteria-for-publication#loc-1), on originality of the presented work.

- Address the concerns by Reviewer 1 on the definition of communication costs in the context of a newly-formed team.

- Justify the methodological aspects highlighted by Reviewer 2 in points 5, 6, 7 and 10. Notice PLOS ONE's publication criterion #3 (https://journals.plos.org/plosone/s/criteria-for-publication#loc-3), on presentation of the methodology.

- Revise interpretations made from the obtained results (especially concerning concerns raised by Reviewer 2 in points 8 and 9 of the report). Notice PLOS ONE's publication criterion #4 (https://journals.plos.org/plosone/s/criteria-for-publication#loc-4), on the link between results and conclusions.

We look forward to receiving your revised manuscript.

Kind regards,

Sergi Lozano

Academic Editor

PLOS ONE

Journal Requirements:

2. Please amend your Methods section and Ethics Statement to state whether participants in the DreamTeam data provided informed consent. Please state the type of consent (e.g., written, verbal, etc).

This work has been supported by the National Science Foundation Awards 560

SES-2021117, SMA-1856090, a 2020 Microsoft Research Dissertation Grant, and 561

National Institutes of Health Awards 1R01GM112938-01, 1R01GM137410-01

Diego Gomez-Zara (DGZ)

Noshir Contractor (NC)

National Science Foundation Award SES-2021117 (DGZ, NC)

https://www.nsf.gov/awardsearch/showAward?AWD_ID=2021117

National Science Foundation Award SMA-1856090

https://www.nsf.gov/awardsearch/showAward?AWD_ID=1856090&HistoricalAwards=false

2020 Microsoft Research Dissertation Grant (DGZ)

https://www.microsoft.com/en-us/research/blog/2020-microsoft-research-dissertation-grant-supports-students-cutting-edge-work/

National Institutes of Health Awards 1R01GM112938-01 (NC) 

https://reporter.nih.gov/search/BVIUPOLACkaltHszAOD0bg/project-details/8802448

National Institutes of Health Awards 1R01GM137410-01 (NC).

https://reporter.nih.gov/search/1gRS-DiRp06XJhDj5VuMFQ/project-details/9981984

4. We noted in your submission details that a portion of your manuscript may have been presented or published elsewhere. [This is an extended and revised version of a preliminary conference article that was 563 presented in Complex Networks 2020.] Please clarify whether this conference proceeding or publication was peer-reviewed and formally published. If this work was previously peer-reviewed and published, in the cover letter please provide the reason that this work does not constitute dual publication and should be included in the current manuscript.

7. Please remove your figures from within your manuscript file, leaving only the individual TIFF/EPS image files, uploaded separately.  These will be automatically included in the reviewers’ PDF.

Reviewers' comments:

Reviewer's Responses to Questions

**Comments to the Author**

1. Is the manuscript technically sound, and do the data support the conclusions?

Reviewer #1: Yes

Reviewer #2: Yes

2. Has the statistical analysis been performed appropriately and rigorously? 

Reviewer #1: N/A

Reviewer #2: I Don't Know

3. Have the authors made all data underlying the findings in their manuscript fully available?

Reviewer #1: No

Reviewer #2: No

4. Is the manuscript presented in an intelligible fashion and written in standard English?

Reviewer #1: Yes

Reviewer #2: Yes

5. Review Comments to the Author

Reviewer #1: The paper focuses on the problem of creating teams with a high diversity of member traits and a high degree of prior relationships. The authors adapt a genetic algorithm to the setting of team formation to achieve teams that satisfy the mentioned criteria and evaluate the algorithm on 2 datasets.

The problem is well motivated (although I have some concerns regarding the communication cost which I have described below) and the algorithm is well explained, and I especially appreciate the pseudo-codes.

I ask that the authors further elaborate on why the communication cost in the problem context (a newly formed team) is defined the way it is. I believe communication cost is important and should be taken into account to address the cold start problem. But if a pair of people in a chosen team aren’t connected by an edge, meaning that they do not already immediately know each other, does it matter if they’re 3 hops away from each other or 4? Especially in the context of workplace or student teams, members of a team typically use computer-mediated communication (CMC) tools to communicate with each other. So while the lack of prior immediate connection or experience with working with a teammate can cause communication problems, it would not entail that a person A whose distance to person B is 3 hops, needs to communicate with B through the 2 persons on his/her route to B and thus would have less of a problem than person C who is connected to B through 4 hops (3 persons), as they both would use the same CMC tools.

In the Discussion, references 60 and 61 are cited in support of the hypothesis that teaming up with people with whom one has indirect relationships can potentially promote familiarity and psychological safety but I did not see this point made in those papers.

I ask that the authors comment on how the algorithm can accommodate traits diversifying which is in fact not desirable (e.g., member preferences for team leadership/hierarchy style need to be homogenous within a team).

In their prior conference paper which this manuscript extends, the authors evaluate the algorithm on 2 courses from the myDreamTeam dataset. Are any of the 3 courses discussed in the current manuscript the ones on which they had evaluated their algorithm before? I ask that the authors more clearly indicate which parts in this manuscript have not been published and discussed before.

Reviewer #2: Thank you for the opportunity to review the manuscript entitled "In search of diverse and connected teams: a computational approach to assemble diverse teams based on members' social networks". The authors addressed an important and interesting question of team formation, and proposed a computational approach to assemble teams with high levels of diversity and familiarity at the same time. I find the explanation of the algorithm clearly communicated. I also enjoyed the visualization of team formation with different tools. That being said, I feel this paper can benefit from further clarification. I list my comments and questions below, and hope the authors find them helpful to improve the manuscript.

Theory

1. I like your focus on maximizing skill diversity and minimizing communication costs. But I find your introduction somewhat perplexing. The first paragraph emphasized on forming teams for optimal team performance; the second paragraph talked about efficiency of team building – finding solutions with minimized time and memory (line 44). Then in line 94 you mentioned there are other objectives such as “minimize communication cost, minimize personnel cost, maximize skills present in a team”, and suddenly summarized your aim of maximizing skill diversity and minimizing communication costs. It confused me. Why did you review other objectives of team formation (and how do they relate to your research concern)? Why did you choose to focus on the combination of skill diversity and communication costs rather than other combinations? And how does your objective link to prior focuses on efficiency and outcomes (or other multi-objective algorithms)? Please clarify.

2. Literature review of team diversity. In the introduction you reviewed that diversity is beneficial for team creativity and innovation (line 8-9). Based on this, you encouraged the diversity of individual attributes (e.g., age, gender, race, and skills, line 179) in team formation. I found this questionable. Prior meta-analytical reviews of team diversity have underscored the contingency perspective in the effectiveness of team diversity. Although functional diversity is often found positive, demographic diversity has no or even negative relationship with team outcomes (Bell, Villado, Lukasik, Belau, & Briggs, 2011). Please extend your literature review, and discuss if it makes sense to maximize skill diversity for all sorts of individual attributes in team formation.

3. Theoretical novelty. Can you clarify your contribution and novelty? Is it about the computational approach you used? Yet as you reviewed, this NSGA-II algorithm was used in prior studies already (Pérez-Toledano, et al., 2019). How is your approach different? Or is it about the new objective you proposed of maximizing diversity and minimizing communication cost (line 445)? It’ll help your readers better grasp your contribution.

4. Theoretical implications. I understand that this algorithm can potentially help practitioners assemble teams for this particular purpose. What I am missing here is the implications for team research. Could you explain and elaborate on your contributions to the team literature. For example, how do your findings advance our understanding of team formation such as the formation process?

Data

5. Choice of your datasets. You tested this algorithm on both My Dream Team Builder and bibsonomy (line 372). Whereas I find My Dream Team Builder a highly relevant and unique sample for your research question, I have trouble understanding why the second dataset of bibsonomy is chosen. First, what are the teams in the bibsonomy dataset? Do you count multi-authored publications as teams? More importantly, why does this dataset qualify for testing your algorithm? You proposed this algorithm to build teams with maximized skill diversity and minimized communication costs. But I am not sure if scientific collaborations shared the same objectives. I find it a bit difficult to envision scientists collaborate to maximize the skill diversity of team projects. Also, My Dream Team Builder creates teams instantly. But in scientific collaboration (or teambuilding), authors may join at different stages of the team project. Should this be a concern as well? Please explain why these two datasets were suitable for testing this algorithm.

Analysis and Results

6. Choice of the algorithms for comparison. I wonder why you decided to compare your algorithm with PLS and SPEA-2. I did not see any of these two methods reviewed in the introduction. Instead, in the introduction, you reviewed single-objective algorithms and mentioned other multi-objective algorithms such as the Multi-objective Particle Swarm Optimization (MOPSO) algorithm (Zhang & Zhang, 2013) and the parallel hybrid grouping genetic algorithm (HGGA, Agustín-Blas, et al., 2011). Could you explain why you assessed PLS and SPEA-2, but not the other multi-objective algorithms that you cited? What motivated you to compare these three?

7. In a relevant vein, I also have trouble understanding why you did not compare with those single-objective algorithms that you spent quite some effort to review (line 94). Logically, it also makes sense to see how your multi-objective algorithm outperforms those single-objective counterparts, such as the MCC algorithm you reviewed (line 101) on minimizing communication cost.

8. Evaluation criteria. You concluded that the NSGA-II algorithm outperforms SPEA-2 and PLS (line 419-420). What are the criteria you based on? Did you rely on the visualization of Figure 2 only? Do you use any quantitative results for this conclusion, such as the product of communication cost of diversity index for each algorithm? please add more details here.

9. Interpretation of Figure 2. I find it a little confusing when I read the figures. First, NSGA-II presents many more solutions (the number of dots in the figures). But the variation across different NSGA-II solutions is also very high. Although NSGA-II solutions tend to relatively locate on the top left area, the large variance on both dimensions is concerning. In contrast, the SPEA-2, PLS, or even random options seem much more concentrated. Does this suggest anything about the reliability of NSGA-II results? How can we interpret the dispersion when evaluating its performance? Again, I feel things can be clarified if you can provide a better explanation of the assessment criteria.

10. The test of memory use. Can you explain why it is important to look that the memory and time usage (p. 426-436)? It does not follow your theory part. Does it relate to the objective of using less memory that some prior studies emphasized (line 44)? But you stated repeatedly that the aims of team building should be maximized skill diversity and minimized communication costs. Also, the difference in memory use (ranging from 1.2 MB to 2.7 MB) does not seem very significant to me. Does it make a big difference in reality? Please clarify why the test is relevant and why this difference is valuable.

REFERENCE

Bell, S. T., Villado, A. J., Lukasik, M. A., Belau, L., & Briggs, A. L. (2011). Getting specific about demographic diversity variable and team performance relationships: A meta-analysis. Journal of management, 37(3), 709-743.

6. PLOS authors have the option to publish the peer review history of their article (what does this mean?). If published, this will include your full peer review and any attached files.

Reviewer #1: No

Reviewer #2: **Yes: **Yingjie Yuan

---

## [Author Response · Author response to Decision Letter 0]

11 Mar 2022

February 28, 2022

Dear editor and reviewers, 

Thank you for providing us with a thorough and constructive review. We found the points identified to be very helpful in revising and improving the paper. We appreciate that the reviewers saw the merits of this manuscript. We are pleased to submit our revised paper, which has been rewritten to address the concerns raised in the initial review. 

R1 raised concerns about the definition of communication costs. In this revision, we provide a more elaborated explanation of why we used this metric and why it is appropriate in the team formation problem. We also ran the team formation implementation using alternative metrics proposed by Lappas et al. (2009), which are in the Supplementary Information. In addition, we calculated the number of hops within team members of the simulated teams to assess the distance among members. We found that most team members are connected directly or through two hops. High-level hops should not be a concern. We included this analysis in the Supplementary Information. In the Discussion section, we now indicate how this team formation problem can accommodate traits that require homogeneity rather than heterogeneity. We also fixed the citations, re-established the MDT case presented in the conference paper, and indicated the extensions and non-published parts of this version compared to the conference paper. 

R2 requested several clarifications and justifications across the paper’s sections. We re-wrote the Introduction section to provide more clarity and state the paper’s contributions. Based on prior studies, we elaborated on the conditions and settings when team diversity can be beneficial. Moreover, we describe prior studies that indicate the benefits of demographic diversity and functional diversity. We acknowledge that team diversity itself is not enough to enhance teams’ outcomes, so we describe the purpose of considering team familiarity in this problem. Then, we establish our team formation problem as theoretically novel since it considers more than one objective function and assigns all individuals into teams. Our paper differs from prior “best-team” and single-objective formulations. We also elaborate on the theoretical implications of this work for team science. In particular, the role of technologies in the team formation process. 

We also justified the choices of datasets and algorithms used in this paper in more detail. In addition to these new justifications, we added a new algorithm based on particle swarm optimization that aligns with the literature review and a different dataset that includes software developer teams formed on GitHub. Regarding the work that we reviewed, we explain why some of these algorithms and problem formulations are incompatible with our proposed optimization problem. While most of these algorithms consider either a single-objective function or forming one single team, our proposal considers multiple objective functions and assigns all available individuals into teams. 

Furthermore, we included quantitative metrics to evaluate the final Pareto front results, which have been highly used in prior studies. We also explain why the variety of solutions (rather than the convergence in a specific set of solutions) is important for multi-objective problems. Finally, we replaced the test of memory with a test of time complexity. 

In the remainder of this document, we address each of the points raised by the reviewers and explain the steps we have taken to address these concerns in the revised manuscript. 

We appreciate the opportunity to revise and resubmit our paper, and we sincerely hope we can share our contribution at PLOS One. Thank you. 

Sincerely,

The Authors

Editor’s meta-review

Clarify the contribution of this article beyond that of the conference paper mentioned in the text (as required by Reviewer 1). Notice PLOS ONE's publication criterion #1, on the originality of the presented work.

In this revised version, we clarify the contributions and extensions of this article beyond the conference paper (Page 4). We also state the theoretical and practical implications of this work in the Introduction and Discussion sections. We confirm that the extended material of this paper has not been published in any previous publications. More details are provided in response to R1 in this letter (Page 5). 

Address the concerns by Reviewer 1 on the definition of communication costs in the context of a newly-formed team.

We have addressed R1’s concerns on the definition of communication costs and its interpretation in the context of team formation. The clarification is in this response letter on page 3 and in the manuscript on page 8.

Justify the methodological aspects highlighted by Reviewer 2 in points 5, 6, 7, and 10. Notice PLOS ONE's publication criterion #3, on the presentation of the methodology.

In this revised version, we have updated and clarified the choices of datasets (Point 5), the choices of algorithms (Point 6), the use of multi-objective functions (Point 7), and the metrics used (Point 10). Beyond the clarifications provided in this new revised version, we: 

Added a new dataset to test our problem implementation (Point 5).

Added a new algorithm based on a hybrid version of particle swarm optimization (HPSO) to test our proposed algorithm (Point 6). 

Added the hypervolume and the unique non-dominated front ratio metrics to provide a quantitative evaluation of the algorithms (Point 10).

We also confirm that we have revised our methodological aspects guided by PLOS ONE’s publication criteria #3. We describe methods and reagents in sufficient detail for another researcher to reproduce the experiments described.

Revise interpretations made from the obtained results (especially concerning concerns raised by Reviewer 2 in points 8 and 9 of the report). Notice PLOS ONE's publication criterion #4, on the link between results and conclusions.

We revised and updated our interpretations made from the obtained results. In particular, we now include quantitative metrics (i.e., Hypervolume, Nondominated Front Size, coverage) to support our results (Point 8) and explain why the variation among results is important for the algorithms (Point 9). We provide more details and justifications on the response to R2 on page 8. We also confirm that we have revised the link between our results and conclusions according to PLOS ONE's publication criterion #4. We also avoid overstating our conclusions. 

Reviewer 1

I ask that the authors further elaborate on why the communication cost in the problem context (a newly formed team) is defined the way it is. I believe communication cost is important and should be taken into account to address the cold start problem. But if a pair of people in a chosen team aren’t connected by an edge, meaning that they do not already immediately know each other, does it matter if they’re 3 hops away from each other or 4? Especially in the context of workplace or student teams, members of a team typically use computer-mediated communication (CMC) tools to communicate with each other. So while the lack of prior immediate connection or experience with working with a teammate can cause communication problems, it would not entail that a person A whose distance to person B is 3 hops, needs to communicate with B through the 2 persons on his/her route to B and thus would have less of a problem than person C who is connected to B through 4 hops (3 persons), as they both would use the same CMC tools.

We use communication cost as a proxy for familiarity among team members: two nodes are connected by an edge if the members have communicated or collaborated before (Kargar et al., 2013). We followed this definition since it has been employed in the team formation algorithms literature to determine collaborations among individuals (Farhadi et al., 2012; Kargar & An, 2011; Lappas et al., 2009). This definition focus on the importance of collaborations and familiarity between experts by taking into account the cost of their collaborations (which Lappas et al. (2009) referred to as communication). According to their model, fewer hops represent lower costs of collaboration. The lowest value possible is when all team members know each other (i.e., they are directly connected), and the highest one is when all members are not connected at all. 

The goal of using the sum of distances as a communication cost metric is to operationalize the number of direct connections and shared connections within a team. While one-hops between team members mean that they already worked together in the past, two-hops mean that members had common prior collaborations in the past. This approach is grounded on triadic closure, which posits that nodes are likely to establish a new connection when they have a connection in common. Three-hops and 4-hops can follow the same principles based on balance theories. For example, if user A has worked with user B, user B has worked with user C, and user C worked with user D, then it is likely for user A (or user B) to establish a new connection with user D. Therefore, using communication cost definition allowed our objective function to search for teams that maximized the number of direct collaborations (i.e. no hops), common connections (one hop), and close connections (two hops and more).

Regarding the context of CMC tools, Lappa et al.’s communication cost definition differs from routing problems. This definition does not require that individuals communicate with each other only through their hops, it only refers to how their prior collaboration networks were constituted. In other words, if user A has a 3-hop distance to user B, it means that they are connected by two people who worked together in the past. It does not mean that user A and user B will communicate through their contacts while they work as a team. This definition focuses on their collaborations rather than their communication channels. 

To understand the effect of three and higher levels hops, we now report the social network structures contained in these databases. We did not provide this information in the initial submission and we added it to this revised submission to provide more context about people’s relationships (Table 2). Overall, individuals were closely related to each other, and the average diameter of these networks (i.e. the longest short path among all members) was 5.3 for the MDT databases, 3.66 for the bibsonomy databases, and 3.0 for the GHTorrent databases. 

We also calculated the frequency of direct contacts, 1-hop, 2-hops, 3-hops in the assembled teams to understand the distances among team members (See S1 Table). As result, we found that the vast majority of the members were connected with others with 2 hops (~31%), followed by direct contacts (~30%). These numbers show that these members were highly connected in general, and a high number of hops was not commonly seen. We hope this information can clarify the minimum effect produced by large hops on the team formation problem. 

We acknowledge that there are many other possible ways to define communication costs among members. One alternative is using weighted graphs in which edges represent how familiar team members are with each other: a low-weight edge implies that members have collaborated a few times, while a high-weight edge implies that members have collaborated multiple times. Other alternatives for communication cost are the diameter (i.e., the largest shortest path between any two nodes in the graph), and the minimum spanning tree (i.e., the minimum sum of the weights of a graph’s edges) (Lappas et al., 2009). We repeated our objective function using these definitions and the results are consistent with the ones reported in the main paper. We have included these alternative definitions in the Supplementary Materials. 

These clarifications, analyses, and observations are now included in our Methods, Results, and Discussion sections.

In the Discussion, references 60 and 61 are cited in support of the hypothesis that teaming up with people with whom one has indirect relationships can potentially promote familiarity and psychological safety but I did not see this point made in those papers.

Thank you for pointing out this issue. We realized that we made a mistake when we added the citations. The original reference was: “Huckman, R. S., Staats, B. R., & Upton, D. M. (2009). Team familiarity, role experience, and performance: Evidence from Indian software services. Management Science, 55(1), 85-100.”, and the reference in the initial submission was “Huberman, B. A., & Hogg, T. (1995). Communities of practice: Performance and evolution. Computational & Mathematical Organization Theory, 1(1), 73-92.” There was an error when we typed “Hu” in the bibliography manager. 

Reference #61 was also not appropriate for supporting the hypothesis that teaming up with familiar members can promote psychological safety. We have addressed this assertion by citing these two papers:

 Staats, B. R., Gino, F., Pisano, G. P., Edmondson, D. H., Pierce, L., & Spektor, E. M. (2010). Varied experience, team familiarity, and learning: The mediating role of psychological safety. 

These authors ran an experimental study in which participants must resolve multiple tasks and demonstrated that “ team familiarity would positively influence psychological safety” (Page 15).

O’Donovan, R., & McAuliffe, E. (2020). Exploring psychological safety in healthcare teams to inform the development of interventions: combining observational, survey and interview data. BMC health services research, 20(1), 1-16.

In their results, the authors found that “familiarity between team members facilitated psychological safety. Team members found it easier to speak openly as they got to know one another better and worked together for longer.” (Page 8).

We have fixed these citations and edited our assertions in our revised manuscript. 

I ask that the authors comment on how the algorithm can accommodate traits diversifying which is in fact not desirable (e.g., member preferences for team leadership/hierarchy style need to be homogenous within a team).

The current algorithm’s implementation supports more than two objective functions. Given the current implementation, we can add a third objective function that minimizes the variance among the team members’ attributes to be considered (e.g., minimize leadership style’s coefficient of variation). This objective function to homogenize members’ attributes can be defined as minimizing the coefficient of variation for continuous variables, or the Blau index for categorical variables. Then, the NSGA-II algorithm will find Pareto fronts that consider (a) lower communication costs, (b) higher diversity in a specific attribute set, and (c) lower diversity in a second attribute set. We added this potential extension in the Discussion section (Page 24). 

In their prior conference paper which this manuscript extends, the authors evaluate the algorithm on 2 courses from the myDreamTeam dataset. Are any of the 3 courses discussed in the current manuscript the ones on which they had evaluated their algorithm before?

Only one course discussed in the current manuscript was evaluated in our prior conference paper. We changed the courses in the current manuscript to test different course levels (undergraduate, graduate, and MBA courses). We put back the missing MDT course from the conference paper to our revised paper. Since this implementation now considers individuals without prior connections (i.e., isolates), the plots and numbers presented in this version are not the same ones presented in our conference paper.

I ask that the authors more clearly indicate which parts in this manuscript have not been published and discussed before.

The parts from this manuscript that have not been published and discussed before are:

The datasets’ generation code and the algorithms’ source code.

The pre-processed and de-identified datasets.

The algorithms’ literature review. We describe the team formation literature in-depth to provide a better motivation and elaborated context for this problem. We include previous algorithms that find one single team and team combinations and algorithms. 

The algorithm’s pseudo-code and explanation of each component step by step. We revised the code and its parts to improve the readability of the code. We provide more detailed descriptions. In contrast, the conference paper provided only the pseudo-code for the crossover step.

This revised version includes an implementation in which :

(a) the number of nodes can be different from a multiple of the team size, leaving a smaller team with the remaining participants; and 

(b) considers members who do not have any edges in the social graph, these members are only considered in the diversity objective function and excluded from the communication cost objective function. 

Evaluation with the bibsonomy dataset and GHTorrent dataset. We included these two datasets to prove that our algorithm can work in other team formation domains. 

Comparison of our implementation against other benchmark algorithms frequently cited in the literature: PLS, SPEA-2, and HPSO. 

We compare the results based on quantitative metrics (hypervolume and the unique non-dominated front ratio) running time used by this algorithm compared to other Pareto-front implementations (PLS, SPEA-2, and HPSO).

We discuss how this implementation can benefit team builders (e.g., managers, instructors) and the consequences of using this team formation algorithm in real teams. 

We also enumerate and make explicit these differences in our revised Introduction section.

Reviewer 2

1. I like your focus on maximizing skill diversity and minimizing communication costs. But I find your introduction somewhat perplexing. The first paragraph emphasized on forming teams for optimal team performance; the second paragraph talked about efficiency of team building – finding solutions with minimized time and memory (line 44). Then in line 94 you mentioned there are other objectives such as “minimize communication cost, minimize personnel cost, maximize skills present in a team”, and suddenly summarized your aim of maximizing skill diversity and minimizing communication costs. It confused me. Why did you review other objectives of team formation (and how do they relate to your research concern)? Why did you choose to focus on the combination of skill diversity and communication costs rather than other combinations? And how does your objective link to prior focuses on efficiency and outcomes (or other multi-objective algorithms)? Please clarify.

We appreciate this observation. We rewrote the Introduction section to provide more clarity and focus on maximizing diversity and familiarity. We removed the mentions of team optimal performance, efficiency in team building, and other objective functions from the Introduction. And we created a new section for our team formation algorithms review. 

We chose these two objective functions because both diversity and familiarity can be determined during the team formation process, and they both determine team composition. Research shows that the interaction of diversity and familiarity can positively influence team performance. While other approaches (e.g., coaching, training, leadership) can help diverse teams work better, those require interventions after teams are assembled. Therefore, we examine how maximizing diversity and familiarity simultaneously can leverage team formation processes and teams’ composition. We elaborated on these reasons in the Introduction section.

Finally, we acknowledge that making strong statements about the “good” or “bad” effects of diversity in teams is a flawed approach (Bell et al., 2011). In this revised version, we explained better when diversity can be beneficial for teams’ efficiency and outcomes, and removed overstatements of the effects of diversity on team performance. In particular, we point out that prior research has shown the benefits of diversity for creativity and innovation. We link our objective goals on the interaction effect between familiarity and diversity on performance (Huckman et al., 2009). These changes are reflected in the new Introduction section.

2. Literature review of team diversity. In the introduction you reviewed that diversity is beneficial for team creativity and innovation (line 8-9). Based on this, you encouraged the diversity of individual attributes (e.g., age, gender, race, and skills, line 179) in team formation. I found this questionable. Prior meta-analytical reviews of team diversity have underscored the contingency perspective in the effectiveness of team diversity. Although functional diversity is often found positive, demographic diversity has no or even negative relationship with team outcomes (Bell, Villado, Lukasik, Belau, & Briggs, 2011). Please extend your literature review, and discuss if it makes sense to maximize skill diversity for all sorts of individual attributes in team formation.

In line 179, we mentioned examples of individuals’ attributes that could be included in the team formation problem. Our formulation emphasizes the use of categorical and numerical variables that can be chosen by the administrator of the algorithm. These attributes can be in the surface-level (e.g., age, race, gender) and deep-level (e.g., backgrounds, careers, functions, expertise) (Harrison et al., 1998). Although we mentioned these attributes as examples, we did not aim to encourage the use of any particular attributes. From our evaluations, the MyDreamTeam dataset was the only one that included demographic variables. We did not include demographic variables in the bibsonomy simulations. And in our third dataset, we only included skills variables. We clarify in the Discussion section that administrators or managers who will administer this algorithm should reflect and decide on adding demographic variables and functional characteristics according to their organizational goals and particular context (Page 25). 

Although prior research on the effects of demographic diversity on team performance has shown mixed results (as we stated in Line 524), several studies have demonstrated their benefits. For example, one study found a positive relationship between gender diversity and team productivity in software engineer teams (Vasilescu et al., 2015), another study found that multi-cultural teams are more likely to provide more creative solutions than teams from a single culture (H.-C. Wang et al., 2011), collective intelligence studies have demonstrated a link between the number of female members and performance (Woolley et al., 2015), and a final example shows that racially diverse teams can compete better than homogeneous teams (Andrevski et al., 2014). Although Bell et al. 2011 found either no relationship between demographic diversity and team performance or small effects, they provide possible explanations and interpretations of these results: (a) how prior studies operationalized race and gender diversity as a variety metric (i.e., number of members of a specific race/sex in the team) rather than separation (i.e., to what extend members seem themselves as a team or not); (b) the context of the study; (c) and the exacerbation of in-group/out-group biases, stereotypes, and prejudices among team members. Overall, there is a consensus that other factors and processes (e.g., familiarity, leadership, perceived diversity, psychological safety, cohesion) moderate the effect of demographic diversity on team performance. Andreviski et al. (2004) found that racial diversity only had a positive effect on team performance when team members had a low aversion against someone from a different race. For these reasons, we consider familiarity as a second objective for team formation because prior relationships in a team can positively moderate the effect of diversity on team performance (Huckman et al., 2009). We added these studies, the advantages and disadvantages of diversity, and the possible effects of moderator variables in the Introduction section.

We also acknowledge that diversity can be beneficial for certain types of tasks only (e.g., creativity tasks, ideation tasks, making-a-decision tasks), which require the combination of different points of view, backgrounds, and experiences (McGrath, 1984). For this reason, we explicitly mention in the Introduction section the benefits of diversity for tasks that require creativity and innovation. We also mention in the Discussion section that the algorithm can also consider optimizing homogenous attributes (e.g., specialization, personality styles) by adding another objective function. 

Lastly, we acknowledge in the Limitations subsection that this approach requires experimentation with real teams to test whether this team formation problem leverages teams’ performance. We have elaborated on these restrictions, scopes, and caveats in the Discussion section.

3. Theoretical novelty. Can you clarify your contribution and novelty? Is it about the computational approach you used? Yet as you reviewed, this NSGA-II algorithm was used in prior studies already (Pérez-Toledano, et al., 2019). How is your approach different? Or is it about the new objective you proposed of maximizing diversity and minimizing communication cost (line 445)? It’ll help your readers better grasp your contribution.

The main contribution of this paper is the formulation of the team formation problem considering teams' diversity levels and members' familiarity simultaneously. As a result, team builders can explore different team combination alternatives, and examine the trade-off between familiarity and diversity. While most studies in team formation algorithms have considered members' skills or personal costs as team formation objective functions (X. Wang et al., 2016), we formulate this optimization problem based on different operationalizations of diversity (i.e., disparity and variety of attributes). This formulation allows choosing various and multiple diversity factors that fit organizational goals (e.g., functional, educational, gender). The second contribution of this work is the design of algorithms for this team formation problem that assigns all available individuals to a team. Previous team formation problems have mainly focused on finding the best team from a pool of individuals and dismissing the rest (Gómez-Zará et al., 2020; X. Wang et al., 2016). This “best-team” approach could not fit organizational goals that require all individuals to belong to a group (e.g., workshops, training classes, location assignment).

We outlined these contributions in the Introduction and Discussion sections. 

4. Theoretical implications. I understand that this algorithm can potentially help practitioners assemble teams for this particular purpose. What I am missing here is the implications for team research. Could you explain and elaborate on your contributions to the team literature. For example, how do your findings advance our understanding of team formation such as the formation process?

The main theoretical implication of this work is the use of computational mechanisms to support team formation processes. Literature has characterized team formation centered on behavioral mechanisms, where teams can be assembled by internal or external forces and based on similarity, familiarity, and competence (Arrow et al., 2000; Hinds et al., 2000). This optimization problem and the implemented algorithms found diverse and connected team combinations that otherwise individuals could not have otherwise foreseen. This work allows team scholars to reflect on the role of technologies in enabling new organizational structures among individuals and organizations, which can lead to new theories of team formation and technologies (Kellogg & Valentine, 2020; Schildt, 2017; Valentine et al., 2017). We elaborate on this implication in the Introduction and Discussion sections.

5. Choice of your datasets. You tested this algorithm on both My Dream Team Builder and bibsonomy (line 372). Whereas I find My Dream Team Builder a highly relevant and unique sample for your research question, I have trouble understanding why the second dataset of bibsonomy is chosen. First, what are the teams in the bibsonomy dataset? Do you count multi-authored publications as teams? More importantly, why does this dataset qualify for testing your algorithm? You proposed this algorithm to build teams with maximized skill diversity and minimized communication costs. But I am not sure if scientific collaborations shared the same objectives. I find it a bit difficult to envision scientists collaborate to maximize the skill diversity of team projects. Also, My Dream Team Builder creates teams instantly. But in scientific collaboration (or teambuilding), authors may join at different stages of the team project. Should this be a concern as well? Please explain why these two datasets were suitable for testing this algorithm.

The purpose of using bibsonomy was only for evaluation purposes. We used this dataset to test the algorithms’ results and running time. A literature review on team formation algorithms shows DBLP, Bibsonomy, IMDB, and GitHub as appropriate examples for evaluation purposes (H.-C. Wang et al., 2011). We chose bibsonomy since some team formation papers tested their algorithms using this database (Anagnostopoulos et al., 2010, 2012). These papers used co-authorship networks as a proxy of relationships and the paper’s topics as authors’ skills. In our example, two authors are connected if they co-authored at least one paper. The skills were calculated based on the papers’ topics. Given that the topics were very broad, we selected the 20 most frequent topics in each journal and computed authors’ skills based on those 20 topics.

In this revision, we include a third database of GitHub repositories provided by GHTorrent. We perform the same exercise assuming that users can create teams based on repositories. Like the previous exercise, the purpose of using this dataset is to demonstrate the algorithms’ capabilities and results.

We acknowledge that forming scientific teams and software teams is complex in reality: new members can be added over time, some specialization is required, not all of them share the same objectives and restrictions, and diversity may be beneficial for certain goals. We believe these restrictions should not be a concern because we use them only to test the algorithms’ efficiency and results. In the revised version of this paper, we acknowledge the restrictions and limitations of these datasets in the Discussion section. 

6. Choice of the algorithms for comparison. I wonder why you decided to compare your algorithm with PLS and SPEA-2. I did not see any of these two methods reviewed in the introduction. Instead, in the introduction, you reviewed single-objective algorithms and mentioned other multi-objective algorithms such as the Multi-objective Particle Swarm Optimization (MOPSO) algorithm (Zhang & Zhang, 2013) and the parallel hybrid grouping genetic algorithm (HGGA, Agustín-Blas, et al., 2011). Could you explain why you assessed PLS and SPEA-2, but not the other multi-objective algorithms that you cited? What motivated you to compare these three?

We recognize that we explained neither the PLS and SPEA-2 algorithms in detail nor why we used them. In this new revision, we add their description and the justification of using them in the Methods section (Pages 16 and 17). Multiple studies have used PLS and SPEA-2 to evaluate and compare multi-objective algorithms (Pérez-Toledano et al., 2019; Zhou et al., 2011; Zihayat et al., 2014). For that reason, we used these algorithms to test our NSGA-II implementation. 

Although we mentioned MOPSO and HGGA to provide examples of algorithms for team formation problems, these algorithms cannot be implemented in our diversity & familiarity team formation problem. Zhang & Zhang’s MOPSO implementation only forms one single team (rather than multiple teams) and it finds the solution in a continuous space. Each solution represents whether a member i belongs to the best team or not. Solutions move in a two-dimensional space, and they apply a sigmoid function to binarize the final outcome. In contrast, our team formation problem represents a combinational problem. Our goal is to assign every available individual to a team and test different team combinations. We operationalize team membership using chromosomes. Therefore, their implementation cannot be used for our particular team formation problem. 

We checked for an alternative MOPSO solution for combinational problems. We found and developed Zhang et al.’s HPSO implementation (2020), which is a hybrid implementation of MOPSO. HPSO combines the particle swarm optimization steps with evolutionary approaches. We decided to use this alternative MOPSO implementation as another benchmark. We explain how this algorithm works in the Methods section and elaborate on its results in the Discussion section. 

Lastly, HGGA cannot be implemented for this particular problem since it was designed for a single-objective problem. 

We acknowledge that our literature review did not follow a clear rationale. In this revised version, we restructured the Related Work to emphasize the focus on previous team formation algorithms and why they cannot be used for our particular optimization problem.

7. In a relevant vein, I also have trouble understanding why you did not compare with those single-objective algorithms that you spent quite some effort to review (line 94). Logically, it also makes sense to see how your multi-objective algorithm outperforms those single-objective counterparts, such as the MCC algorithm you reviewed (line 101) on minimizing communication cost.

The problem with single-objective algorithms is that they only provide one single team. Since they do not implement dominance criteria, the single-objective algorithm will prioritize the best team on one dimension given specific restrictions or constraints. Our goal with this multi-objective implementation is to help team builders examine team combinations with different trade-offs. 

Regarding the MCC algorithm, it aims to find the best team possible given a pool of individuals. It will only provide a team of size n, dismissing the other individuals. In contrast, our approach teams up all the individuals of the pool. We could implement their specific problem using an evolutionary algorithm but it would only provide one single solution. If that is the case, we expect that an MCC implementation will find a combination located in one of the extremes of the Pareto Front if the trade-off parameter is 0 or 1 (i.e., highest diversity, lowest communication cost), or in the middle of the Pareto Front if the tradeoff parameter is 0.5. We provide this clarification in the Related Work section (Pages 6 and 7). 

8. Evaluation criteria. You concluded that the NSGA-II algorithm outperforms SPEA-2 and PLS (line 419-420). What are the criteria you based on? Did you rely on the visualization of Figure 2 only? Do you use any quantitative results for this conclusion, such as the product of communication cost of diversity index for each algorithm? please add more details here.

Thank you for pointing out this. We relied on the visualization of Figure 2 and the time/resources tests in the original version. In this revised version, we include quantitative metrics (i.e., Hypervolume, Nondominated Front Size, coverage) to support our results. The metrics are explained in the Methods section (Page 18), and their results are discussed in the Results section (Pages 19-21).

9. Interpretation of Figure 2. I find it a little confusing when I read the figures. First, NSGA-II presents many more solutions (the number of dots in the figures). But the variation across different NSGA-II solutions is also very high. Although NSGA-II solutions tend to relatively locate on the top left area, the large variance on both dimensions is concerning. In contrast, the SPEA-2, PLS, or even random options seem much more concentrated. Does this suggest anything about the reliability of NSGA-II results? How can we interpret the dispersion when evaluating its performance? Again, I feel things can be clarified if you can provide a better explanation of the assessment criteria.

In this revision, we provide a better explanation of the assessment criteria and the importance of diverse solutions (Cao et al., 2015). Pareto Fronts’ shapes provide useful information about the amount of tradeoff between different dimensions (e.g., communication cost, diversity), and how much compromise is needed from some criteria to improve others. Finding the true Pareto front of this team formation problem is computationally hard given we need to compute and assess all the possible team combinations. For this reason, algorithms use a series of steps to find an approximation of the true Pareto front. A critical assumption for these algorithms is that the Pareto Front is sufficiently populated. The quality of this approximation depends upon (1) the proximity of the points on the approximated front to the points on the true Pareto front; and (2) the diversity of the solutions on the approximated front, where more diversity is typically better. Although the true Pareto front is unknown, solutions that dominate others are close to the theoretical true Pareto Front. Then, the diversity of the solutions will provide a larger range and granularity of the Pareto Front. 

That said, the large variance on both dimensions shows that the NSGA-II algorithm found more non-dominated solutions, which is desirable and not concerning. The crowding distance step of NSGA-II allowed the algorithm to keep a broader range of non-dominant solutions. Additionally, the algorithm kept secondary solutions in different layers that could have originated non-dominant solutions in later iterations. As the algorithm continues creating new generations, non-dominant solutions can be still considered to find other potential solutions. Moreover, NSGA-II can still identify non-dominant solutions in the middle of the trade-off. 

In contrast, the low variance of the other algorithms shows that they converged on a specific set of solutions and to a specific trade-off. They do not consider other possible combinations that prioritize either familiarity or diversity. Therefore, these algorithms can lack diverse solutions that reside in the extremes of both dimensions. 

We have articulated a better explanation of a Pareto front in the Related Work section (Page 6) and the interpretations of the results in the Result section. 

10. The test of memory use. Can you explain why it is important to look that the memory and time usage (p. 426-436)? It does not follow your theory part. Does it relate to the objective of using less memory that some prior studies emphasized (line 44)? But you stated repeatedly that the aims of team building should be maximized skill diversity and minimized communication costs. Also, the difference in memory use (ranging from 1.2 MB to 2.7 MB) does not seem very significant to me. Does it make a big difference in reality? Please clarify why the test is relevant and why this difference is valuable.

We recognize that these metrics were not theoretically driven and elaborated without enough details. In this new version, we removed the memory analysis and explain the importance of checking algorithms’ running-time based on computational complexity. 

As the results show, the running times of PLS and HPSO increase exponentially as the input size increases. In contrast, NSGA-II provides solutions faster than the other algorithms, and NSGA-II’s time increase stays at a quadratic scale (O(n^2)). Moreover, NSGA-II required less than one-third of the time that PLS and HPSO took to provide similar results. Therefore, using NSGA-II is highly encouraged to find solutions efficiently as the input size increases. These differences should be considered in addition to the Pareto Front metrics discussed in the revised manuscript. We included these clarifications and interpretations in our Results and Discussion sections. 

References

Anagnostopoulos, A., Becchetti, L., Castillo, C., Gionis, A., & Leonardi, S. (2010). Power in unity: forming teams in large-scale community systems. ACM. https://doi.org/10.1145/1871437.1871515

Anagnostopoulos, A., Becchetti, L., Castillo, C., Gionis, A., & Leonardi, S. (2012). Online team formation in social networks. Proceedings of the 21st International Conference on World Wide Web, 839–848. https://doi.org/10.1145/2187836.2187950

Andrevski, G., Richard, O. C., Shaw, J. D., & Ferrier, W. J. (2014). Racial Diversity and Firm Performance: The Mediating Role of Competitive Intensity. Journal of Management, 40(3), 820–844. https://doi.org/10.1177/0149206311424318

Arrow, H., McGrath, J. E., & Berdahl, J. L. (2000). Small groups as complex systems: Formation, coordination, development, and adaptation. Sage Publications.

Backstrom, L., Boldi, P., Rosa, M., Ugander, J., & Vigna, S. (2012). Four degrees of separation. Proceedings of the 4th Annual ACM Web Science Conference, 33–42. https://doi.org/10.1145/2380718.2380723

Bell, S. T., Villado, A. J., Lukasik, M. A., Belau, L., & Briggs, A. L. (2011). Getting Specific about Demographic Diversity Variable and Team Performance Relationships: A Meta-Analysis. Journal of Management, 37(3), 709–743. https://doi.org/10.1177/0149206310365001

Cao, Y., Smucker, B. J., & Robinson, T. J. (2015). On using the hypervolume indicator to compare Pareto fronts: Applications to multi-criteria optimal experimental design. Journal of Statistical Planning and Inference, 160, 60–74. https://doi.org/10.1016/j.jspi.2014.12.004

Contractor, N. (2001). Small worlds: The dynamics of networks between order and randomness. Chance, 14(4), 33–36.

Farhadi, F., Hoseini, E., Hashemi, S., & Hamzeh, A. (2012). TeamFinder: A Co-clustering based Framework for Finding an Effective Team of Experts in Social Networks. 2012 IEEE 12th International Conference on Data Mining Workshops, 107–114. https://doi.org/10.1109/ICDMW.2012.54

Gómez-Zará, D., DeChurch, L. A., & Contractor, N. S. (2020). A Taxonomy of Team-Assembly Systems: Understanding How People Use Technologies to Form Teams. Proceedings of the ACM on Human-Computer Interaction, 2(CSCW2), 36. https://doi.org/10.1145/3415252

Harrison, D. A., Price, K. H., & Bell, M. P. (1998). Beyond relational demography: Time and the effects of surface-and deep-level diversity on work group cohesion. Academy of Management Journal. Academy of Management. https://journals.aom.org/doi/abs/10.5465/256901

Hinds, P. J., Carley, K. M., Krackhardt, D., & Wholey, D. (2000). Choosing work group members: Balancing similarity, competence, and familiarity. Organizational Behavior and Human Decision Processes, 81(2), 226–251.

Huckman, R. S., Staats, B. R., & Upton, D. M. (2009). Team Familiarity, Role Experience, and Performance: Evidence from Indian Software Services. Management Science, 55(1), 85–100. https://doi.org/10.1287/mnsc.1080.0921

Kargar, M., & An, A. (2011). TeamExp: Top-k Team Formation in Social Networks. 2011 IEEE 11th International Conference on Data Mining Workshops, 1231–1234. https://doi.org/10.1109/ICDMW.2011.162

Kargar, M., Zihayat, M., & An, A. (2013). Finding Affordable and Collaborative Teams from a Network of Experts. In Proceedings of the 2013 SIAM International Conference on Data Mining (SDM) (pp. 587–595). Society for Industrial and Applied Mathematics. https://doi.org/10.1137/1.9781611972832.65

Kellogg, K. C., & Valentine, M. A. (2020). Algorithms at work: The new contested terrain of control. Academy of Management. https://journals.aom.org/doi/abs/10.5465/annals.2018.0174

Lappas, T., Liu, K., & Terzi, E. (2009). Finding a team of experts in social networks. Proceedings of the 15th ACM SIGKDD International Conference on Knowledge Discovery and Data Mining, 467–476. https://doi.org/10.1145/1557019.1557074

Milgram, S. (1967). The small world problem. Psychology Today, 2(1), 60–67. http://files.diario-de-bordo-redes-conecti.webnode.com/200000013-211982212c/AN%20EXPERIMENTAL%20STUDY%20by%20Travers%20and%20Milgram.pdf

Pérez-Toledano, M. Á., Rodriguez, F. J., García-Rubio, J., & Ibañez, S. J. (2019). Players’ selection for basketball teams, through Performance Index Rating, using multiobjective evolutionary algorithms. PloS One, 14(9), e0221258. https://doi.org/10.1371/journal.pone.0221258

Schildt, H. (2017). Big data and organizational design – the brave new world of algorithmic management and computer augmented transparency. Innovation (North Sydney, N.S.W.), 19(1), 23–30. https://doi.org/10.1080/14479338.2016.1252043

Uzzi, B., Amaral, L. A. N., & Reed-Tsochas, F. (2007). Small-world networks and management science research: a review. European Management Review, 4(2), 77–91. https://doi.org/10.1057/palgrave.emr.1500078

Valentine, M. A., Retelny, D., To, A., Rahmati, N., Doshi, T., & Bernstein, M. S. (2017). Flash Organizations: Crowdsourcing Complex Work by Structuring Crowds As Organizations. In Proceedings of the 2017 CHI Conference on Human Factors in Computing Systems (pp. 3523–3537). Association for Computing Machinery. https://doi.org/10.1145/3025453.3025811

Vasilescu, B., Posnett, D., Ray, B., van den Brand, M. G. J., Serebrenik, A., Devanbu, P., & Filkov, V. (2015). Gender and Tenure Diversity in GitHub Teams. 3789–3798. https://doi.org/10.1145/2702123.2702549

Wang, H.-C., Fussell, S. R., & Cosley, D. (2011). From Diversity to Creativity: Stimulating Group Brainstorming with Cultural Differences and Conversationally-retrieved Pictures. Proceedings of the ACM 2011 Conference on Computer Supported Cooperative Work, 265–274. https://doi.org/10.1145/1958824.1958864

Wang, X., Zhao, Z., & Ng, W. (2016). USTF: A Unified System of Team Formation. IEEE Transactions on Big Data, 2(1), 70–84. https://doi.org/10.1109/TBDATA.2016.2546303

Woolley, A. W., Aggarwal, I., & Malone, T. W. (2015). Collective Intelligence and Group Performance. Current Directions in Psychological Science, 24(6), 420–424. https://doi.org/10.1177/0963721415599543

Zhang, X., Guo, P., Zhang, H., & Yao, J. (2020). Hybrid Particle Swarm Optimization Algorithm for Process Planning. Science in China, Series A: Mathematics, 8(10), 1745. https://doi.org/10.3390/math8101745

Zhou, A., Qu, B.-Y., Li, H., Zhao, S.-Z., Suganthan, P. N., & Zhang, Q. (2011). Multiobjective evolutionary algorithms: A survey of the state of the art. Swarm and Evolutionary Computation, 1(1), 32–49. https://doi.org/10.1016/j.swevo.2011.03.001

Zihayat, M., Kargar, M., & An, A. (2014). Two-Phase Pareto Set Discovery for Team Formation in Social Networks. 2014 IEEE/WIC/ACM International Joint Conferences on Web Intelligence (WI) and Intelligent Agent Technologies (IAT), 2, 304–311. https://doi.org/10.1109/WI-IAT.2014.112

---

## [Decision Letter · Decision Letter 1]

15 Jul 2022

PONE-D-21-21409R1In search of diverse and connected teams: a computational approach to assemble diverse teams based on members' social networksPLOS ONE

Dear Dr. Gomez-Zara,

Thank you for submitting your manuscript to PLOS ONE. After careful consideration, we feel that it has merit but does not fully meet PLOS ONE’s publication criteria as it currently stands. Therefore, we invite you to submit a revised version of the manuscript that addresses the points raised during the review process. As you can see below, both reviewers have recommended publication. However, Reviewer #1 has asked for some intuition on the quality metrics introduced, and has pointed out some minor issues. Please, address them in your revised version. Moreover, since PLOS ONE does not proof-read manuscript, I suggest revising the text for small typos and shortcomings. For instance, the word 'otherwise' is written twice in line 979.

We look forward to receiving your revised manuscript.

Kind regards,

Sergi Lozano

Academic Editor

PLOS ONE

Journal Requirements:

Reviewers' comments:

Reviewer's Responses to Questions

**Comments to the Author**

1. If the authors have adequately addressed your comments raised in a previous round of review and you feel that this manuscript is now acceptable for publication, you may indicate that here to bypass the “Comments to the Author” section, enter your conflict of interest statement in the “Confidential to Editor” section, and submit your "Accept" recommendation.

Reviewer #1: All comments have been addressed

Reviewer #3: All comments have been addressed

2. Is the manuscript technically sound, and do the data support the conclusions?

Reviewer #1: Yes

Reviewer #3: Yes

3. Has the statistical analysis been performed appropriately and rigorously? 

Reviewer #1: N/A

Reviewer #3: Yes

4. Have the authors made all data underlying the findings in their manuscript fully available?

Reviewer #1: Yes

Reviewer #3: Yes

5. Is the manuscript presented in an intelligible fashion and written in standard English?

Reviewer #1: Yes

Reviewer #3: Yes

6. Review Comments to the Author

Reviewer #1: Thank you for responding to the reviews. I find the approach, the choice of the algorithm, and the communication cost generally better motivated compared to the initial submission. The new Related Work section is now also better connected to the rest of the paper.

I appreciate the authors clarifying the points that I had asked for including the definition of communication cost as well as clarifying the average network diameter and frequency of direct contacts.

One request I have is for the authors is to provide intuition about the metrics that they have used for evaluating the NSGA-II algorithm and why they matter in the specific context of team formation. For example, intuitively, why is a higher hypervolume value preferred and what qualities about a selection of teams does it demonstrate?

Minor points:

Line 318-319 is awkwardly phrased and I had a hard time reading it at first. The clause “and it has been classified as an NP-hard problem” in effect refers to the minimization problem but with the sentence it is conjoined with, it seems as if it refers to computing the communication cost which in fact, can be done in polynomial time.

Line 296: P_j & P_j => P_i & P_j

Reviewer #3: The authors have approached an interesting topic in a novel way, pairing group dynamics with computer science. The methods and analyses are sound, and the results are intriguing. I would support publication of this manuscript in its current form.

7. PLOS authors have the option to publish the peer review history of their article (what does this mean?). If published, this will include your full peer review and any attached files.

Reviewer #1: No

Reviewer #3: No

---

## [Author Response · Author response to Decision Letter 1]

8 Sep 2022

September 9th, 2022

Dear Dr. Lozano and reviewers, 

We thank you and appreciate your precious time and efforts in revising our manuscript. Your valuable and insightful observations allowed us to improve this article. We are pleased to submit our revised paper to address the most recent comments. 

R1 requested providing more intuition about the metrics used to evaluate the algorithms. We added more context and rationale in the Metrics sub-section and Results section. Also, we have fixed the typos identified by R1. In addition, two authors have proofread this new version and fixed typos and grammar errors.

We also want to thank R3 for the kind comments and for taking the time to revise this resubmission. 

In the remainder of this document, we address the points raised in the last review and explain the steps we have taken to address these concerns in this revised manuscript. 

We appreciate the opportunity to revise and resubmit our paper, and we sincerely hope we can share our contribution at PLOS One. Thank you. 

Sincerely,

The Authors

---------

Editor’s meta-review

As you can see below, both reviewers have recommended publication. However, Reviewer #1 has asked for some intuition on the quality metrics introduced, and has pointed out some minor issues. Please, address them in your revised version. 

>> In this revised version, we elaborate on the rationale of the metrics used in this study. More details are provided in response to R1 in this letter. 

Moreover, since PLOS ONE does not proof-read manuscript, I suggest revising the text for small typos and shortcomings. For instance, the word 'otherwise' is written twice in line 979.

>> Two authors have proofread this new version and fixed typos and grammar errors. 

-----------------

Reviewer 1

One request I have is for the authors is to provide intuition about the metrics that they have used for evaluating the NSGA-II algorithm and why they matter in the specific context of team formation. For example, intuitively, why is a higher hypervolume value preferred and what qualities about a selection of teams does it demonstrate?

>> In this revised version, we provide more insight into the metrics used by this study. In summary, higher hypervolume scores show that teams with higher levels of diversity and familiarity can be found. The Unique Non-dominated Front Ratio shows how many high-diversity/high-familiarity team combinations were found by only one algorithm. Lastly, evaluating the running time is essential since some problems will require large participant pools to be assigned to teams. We develop the rationale and intuition of these metrics in the Metrics sub-section and Results section. 

Line 318-319 is awkwardly phrased and I had a hard time reading it at first. The clause “and it has been classified as an NP-hard problem” in effect refers to the minimization problem but with the sentence it is conjoined with, it seems as if it refers to computing the communication cost which in fact, can be done in polynomial time.

>> Thank you for pointing out this error. It was in fact referring to the minimization problem and not to computing the communication cost. We removed the sentence and the resulting paragraph is the following: “The goal is to minimize the average sum of shortest path lengths across all assembled teams in the individuals’ network. Computing the sum of communication costs of a set of teams runs in O(n2) time.”

Line 296: P_j & P_j => P_i & P_j

>> Fixed.

---

## [Decision Letter · Decision Letter 2]

28 Sep 2022

In search of diverse and connected teams: a computational approach to assemble diverse teams based on members' social networks

PONE-D-21-21409R2

Dear Dr. Gomez-Zara,

We’re pleased to inform you that your manuscript has been judged scientifically suitable for publication and will be formally accepted for publication once it meets all outstanding technical requirements.

Kind regards,

Seyedali Mirjalili

Academic Editor

PLOS ONE

Additional Editor Comments (optional):

Reviewers' comments:

Reviewer's Responses to Questions

**Comments to the Author**

1. If the authors have adequately addressed your comments raised in a previous round of review and you feel that this manuscript is now acceptable for publication, you may indicate that here to bypass the “Comments to the Author” section, enter your conflict of interest statement in the “Confidential to Editor” section, and submit your "Accept" recommendation.

Reviewer #1: All comments have been addressed

2. Is the manuscript technically sound, and do the data support the conclusions?

Reviewer #1: Yes

3. Has the statistical analysis been performed appropriately and rigorously? 

Reviewer #1: N/A

4. Have the authors made all data underlying the findings in their manuscript fully available?

Reviewer #1: Yes

5. Is the manuscript presented in an intelligible fashion and written in standard English?

Reviewer #1: Yes

6. Review Comments to the Author

Reviewer #1: Thank you for clarifying the intuition behind the metrics used for evaluating the algorithm, and for making edits throughout the paper to ensure better readability. I am glad to recommend the paper for acceptance.

7. PLOS authors have the option to publish the peer review history of their article (what does this mean?). If published, this will include your full peer review and any attached files.

Reviewer #1: No
